# Understanding How Consistency Works in Federated Learning via Stage-wise Relaxed Initialization

**Yan Sun**
The University of Sydney
ysun9899@uni.sydney.edu.au

**Li Shen**[*]
JD Explore Academy
mathshenli@gmail.com

**Dacheng Tao**
The University of Sydney
dacheng.tao@gmail.com

## Abstract

Federated learning (FL) is a distributed paradigm that coordinates massive local clients to collaboratively train a global model via stage-wise local training processes on the heterogeneous dataset. Previous works have implicitly studied that FL suffers from the "client-drift" problem, which is caused by the inconsistent optimum across local clients. However, till now it still lacks solid theoretical analysis to explain the impact of this local inconsistency. To alleviate the negative impact of the "client drift" and explore its substance in FL, in this paper, we first design an efficient FL algorithm *FedInit*, which allows employing the personalized relaxed initialization state at the beginning of each local training stage. Specifically, *FedInit* initializes the local state by moving away from the current global state towards the reverse direction of the latest local state. This relaxed initialization helps to revise the local divergence and enhance the local consistency level. Moreover, to further understand how inconsistency disrupts performance in FL, we introduce the excess risk analysis and study the divergence term to investigate the test error of the proposed *FedInit* method. Our studies show that optimization error is not sensitive to this local inconsistency, while it mainly affects the generalization error bound in *FedInit*. Extensive experiments are conducted to validate this conclusion. Our proposed *FedInit* could achieve state-of-the-art (SOTA) results compared to several advanced benchmarks without any additional costs. Meanwhile, stage-wise relaxed initialization could also be incorporated into the current advanced algorithms to achieve higher performance in the FL paradigm.

## 1 Introduction

Since McMahan et al. [26] developed federated learning, it becomes a promising paradigm to effectively make full use of the computational ability of massive edge devices. Kairouz et al. [17] further classify the modes based on the specific tasks and different environmental setups. Different from centralized training, FL utilizes a central server to coordinate the clients to perform several local training stages and aggregate local models as one global model. However, due to the heterogeneous dataset, it still suffers from significant performance degradation in practical scenarios.

Several previous studies explore the essence of performance limitations in FL and summarize it as the "client-drift" problem [1, 19, 22, 38, 44, 46, 48]. From the perspective of the global target, Karimireddy et al. [19] claim that the aggregated local optimum is far away from the global optimum due to the heterogeneity of the local dataset, which introduces the "client-drift" in FL. However, under limited local training steps, local clients can not genuinely approach the local optimum. To describe this negative impact more accurately, Acar et al. [1] and Wang et al. [44] point out that each locally optimized objective should be regularized to be aligned with the global objective. Moreover, beyond the guarantees of local consistent objective, Xu et al. [46] indicate that the performance

---

[*]Li Shen is the corresponding author.

37th Conference on Neural Information Processing Systems (NeurIPS 2023).

degradation could be further eliminated in FL if it guarantees the local consistent updates at each communication round, which is more similar to the centralized scenarios. These arguments intuitively provide forward-looking guidance for improving the performance in FL. However, in the existing analysis, there is still no solid theoretical support to understand the impact of the consistency term, which also severely hinders the further development of the FL paradigm.

To alleviate the negative impact of the "client-drift" problem and strengthen consistency in the FL paradigm, in this paper, we take into account adopting the personalized relaxed initialization at the beginning of each communication round, dubbed *FedInit* method. Specifically, *FedInit* initializes the selected local state by moving away from the current global state towards the reverse direction of the current latest local state. Personalized relaxed initialization helps each local model to revise its divergence and gather together with each other during the local training process. This flexible approach is surprisingly effective in FL and only adopts a constant coefficient to control the divergence level of the initialization. It could also be easily incorporated as a plug-in into other advanced benchmarks to further improve their performance.

Moreover, to explicitly understand how local inconsistency disrupts performance, we introduce the excess risk analysis to investigate the test error of *FedInit* under the smooth non-convex objective, which includes an optimization error bound and a generalization error bound. Our theoretical studies indicate that the optimization error is insensitive to local inconsistency, while it mainly affects the generalization performance. Under *PŁ*-condition, consistency performs as the dominant term in the excess risk. Extensive empirical studies are conducted to validate the efficiency of the *FedInit* method. On the CIFAR-10/100 dataset, it could achieve SOTA results compared to several advanced benchmarks without additional costs. It also helps to enhance the consistency level in FL.

In summary, the main contributions of this work are stated as follows:

- We propose an efficient and novel FL method, dubbed *FedInit*, which adopts the personalized relaxed initialization state on the selected local clients at each communication round. Relaxed initialization is dedicated to enhancing local consistency during training, and it is also a practical plug-in that could easily to incorporated into other methods.

- One important contribution is that we introduce the excess risk analysis in the proposed *FedInit* method to understand the intrinsic impact of local consistency. Our theoretical studies prove that the optimization error is insensitive to consistency, while it mainly affects the test error and generalization error bound.

- Extensive numerical studies are conducted on the real-world dataset to validate the efficiency of the *FedInit* method, which outperforms several SOTA benchmarks without additional training costs. Meanwhile, as an efficient plug-in, relaxed initialization (*FedInit*) could also help the other benchmarks in our paper to achieve higher performance with effortlessness.

## 2 Related Work

**Consistency in FL.** FL employs an enormous number of edge devices to jointly train a single model among the isolated heterogeneous dataset [17, 26]. As a standard benchmark, *FedAvg* [2, 26, 48] allows the local stochastic gradient descent (local SGD) [10, 23, 45] based updates and uniformly selected partial clients' participation to alleviate the communication bottleneck. The stage-wise local training processes lead to significant divergence for each client [5, 25, 43, 44]. To improve the efficiency of the FL paradigm, a series of methods are proposed. Karimireddy et al. [19] indicate that inconsistent local optimums cause the severe "client drift" problem and propose the *SCAFFOLD* method which adopts the variance reduction [6, 16] technique to mitigate it. Li et al. [22] penalize the prox-term on the local objective to force the local update towards both the local optimum and the last global state. Zhang et al. [49] utilize the primal-dual method to improve consistency via solving local objectives under the equality constraint. Specifically, a series of works further adopt the alternating direction method of multipliers (ADMM) to optimize the global objective [1, 9, 41, 52], which could also enhance the consistency term. Beyond these, a series of momentum-based methods are proposed to strengthen local consistency. Wang et al. [42] study a global momentum update method to stabilize the global model. Further, Gao et al. [8] use a local drift correction via a momentum-based term to revise the local gradient, efficiently reducing inconsistency. Ozfatura et al. [28], Xu et al. [46], Sun et al. [39] propose a similar client-level momentum to force the local update towards the last global direction. A variant of client-level momentum that adopts the inertial momentum to further improve

the local consistency level [24, 40]. At present, improving the consistency in FL remains a very important and promising research direction. Though these studies involve the heuristic discussion on consistency, in this paper we focus on the personalized relaxed initialization.

**Generalization in FL.** A lot of works have studied the properties of generalization in FL. Based on the margin loss [3, 7, 27], Reisizadeh et al. [31] develop a robust FL paradigm to alleviate the distribution shifts across the heterogeneous clients. Shi et al. [32] study the efficient and stable model technique of model ensembling. Yagli et al. [47] prove the information-theoretic bounds on the generalization error and privacy leakage in the general FL paradigm. Qu et al. [29] propose to adopt the sharpness aware minimization (SAM) optimizer on the local client to improve the flatness of the loss landscape. Caldarola et al. [4], Sun et al. [37, 38], Shi et al. [33, 34] propose several variants based on SAM that could achieve higher performance. However, these works only focus on the generalization efficiency in FL, while in this paper we prove that its generalization error bound is dominated by consistency.

## 3 Methodology

### 3.1 Preliminaries

Under the cross-device FL setups, there are a very large number of local clients to collaboratively train a global model. Due to privacy protection and unreliable network bandwidth, only a fraction of devices are open-accessed at any one time [17, 29]. Therefore, we define each client stores a private dataset $\mathcal{S}_i = \{z_j\}$ where $z_j$ is drawn from an unknown unique distribution $\mathcal{D}_i$. The whole local clients constitute a set $\mathcal{C} = \{i\}$ where $i$ is the index of each local client and $|\mathcal{C}| = C$. Actually, in the training process, we expect to approach the optimum of the population risk $F$:

$$w_{\mathcal{D}}^{\star} \in \arg\min_{w} \left\{ F(w) \triangleq \frac{1}{C} \sum_{i \in \mathcal{C}} F_i(w) \right\}, \tag{1}$$

where $F_i(w) = \mathbb{E}_{z_j \sim \mathcal{D}_i} F_i(w, z_j)$ is the local population risk. While in practice, we usually consider the empirical risk minimization of the non-convex finite-sum problem in FL as:

$$w^{\star} \in \arg\min_{w} \left\{ f(w) \triangleq \frac{1}{C} \sum_{i \in \mathcal{C}} f_i(w) \right\}, \tag{2}$$

where $f_i(w) = \frac{1}{S_i} \sum_{z_j \in \mathcal{S}_i} f_i(w; z_j)$ is the local empirical risk. In Section 4.1, we will analyze the difference between these two results. Furthermore, we introduce the excess risk analysis to upper bound the test error and further understand how consistency works in the FL paradigm.

### 3.2 Personalized Relaxed Initialization

In this part, we introduce the relaxed initialization in *FedInit* method. *FedAvg* proposes the local-SGD-based implementation in the FL paradigm with a partial participation selection. It allows uniformly selecting a subset of clients $\mathcal{N}$ to participate in the current training. In each round, it initializes the local model as the last global model. Therefore, after each round, the local models are always far away from each other. The local offset $w_{i,K}^{t-1} - w^t$ is the main culprit leading to inconsistency. Moreover, for different clients, their impacts vary with local heterogeneity. To alleviate this divergence, we propose the *FedInit* method which adopts the personalized relaxed initialization at the beginning of each round. Concretely, on the selected active clients, it begins the local training from a new personalized state, which moves away from the last global model towards the reverse direction from the latest local state (Line.6 in

---

**Algorithm 1:** *FedInit* Algorithm

**Input:** model $w$, local model $w_i, T, K, \beta$.
**Output:** model $w^T$.

1 **Initialize states**: initialize $w^{-1} = w_{i,0}^{-1} = w^0$.
2 **for** $t = 0, 1, ..., T - 1$ **do**
3      randomly select active clients set $\mathcal{N}$ from $\mathcal{C}$
4      **for** $i \in \mathcal{N}$ ***in parallel*** **do**
5          send the $w^t$ to the active clients
6          set the $w^t + \beta(w^t - w_{i,K}^{t-1})$ as $w_{i,0}^t$
7          **for** $k = 0, 1, ..., K - 1$ **do**
8              compute gradient $g_{i,k}^t$ at $w_{i,k}^t$
9              $w_{i,k+1}^t = w_{i,k}^t - \eta g_{i,k}^t$
10          **end**
11          send the $w_i^t = w_{i,K}^t$ to the server
12      **end**
13      $w^{t+1} = \frac{1}{N} \sum_{i \in \mathcal{N}} w_i^t$
14 **end**

---

Algorithm 1). A coefficient $\beta$ is adopted to control the level of personality. This offset $\beta(w^t - w_{i,K}^{t-1})$ in the relaxed initialization (RI) provides a correction that could help local models gather together after the local training process. Furthermore, this relaxed initialization is irrelevant to the local optimizer, which means, it could be easily incorporated into other methods. Additionally, *FedInit* does not require extra auxiliary information to communicate. It is a practical technique in FL.

## 4 Theoretical Analysis

In this section, we first introduce the excess risk in FL which could provide a comprehensive analysis on the joint performance of both optimization and generalization. In the second part, we introduce the main assumptions adopted in our proofs and discuss them in different situations. Then we state the main theorems on the analysis of the excess risk of our proposed *FedInit* method.

### 4.1 Excess Risk Error in FL

Since Karimireddy et al. [19] pointed out that client-drift problem may seriously damage the performance in the FL paradigm, many previous works [15, 18, 19, 30, 36, 44, 46, 48] have learned its inefficiency in the FL paradigm. However, most of the analyses focus on the studies from the onefold perspective of optimization convergence but ignore investigating its impact on generality. To further provide a comprehensive understanding of how client-drift affects the performance in FL, we adopt the well-known excess risk in the analysis of our proposed *FedInit* method.

We denote $w^T$ as the model generated by *FedInit* method after $T$ communication rounds. Compared with $f(w^T)$, we mainly focus on the efficiency of $F(w^T)$ which corresponds to its generalization performance. Therefore, we analyze the $\mathbb{E}[F(w^T)]$ from the excess risk $\mathcal{E}_E$ as:

$$\mathcal{E}_E = \mathbb{E}[F(w^T)] - \mathbb{E}[f(w^*)] = \underbrace{\mathbb{E}[F(w^T) - f(w^T)]}_{\mathcal{E}_G:\ generalization\ error} + \underbrace{\mathbb{E}[f(w^T) - f(w^*)]}_{\mathcal{E}_O:\ optimization\ error}. \quad (3)$$

Generally, the $\mathbb{E}[f(w^*)]$ is expected to be very small and even to zero if the model could well-fit the dataset. Thus $\mathcal{E}_E$ could be considered as the joint efficiency of the generated model $w^T$. Thereinto, $\mathcal{E}_G$ means the different performance of $w^T$ between the training dataset and the test dataset, and $\mathcal{E}_O$ means the similarity between $w^T$ and optimization optimum $w^*$ on the training dataset.

### 4.2 Assumptions

In this part, we introduce some assumptions adopted in our analysis. We will discuss their properties and distinguish the proofs they are used in.

**Assumption 1** *For $\forall w_1, w_2 \in \mathbb{R}^d$, the non-convex local function $f_i$ satisfies $L$-smooth if:*

$$\|\nabla f_i(w_1) - \nabla f_i(w_2)\| \le L\|w_1 - w_2\|. \quad (4)$$

**Assumption 2** *For $\forall w \in \mathbb{R}^d$, the stochastic gradient is bounded by its expectation and variance as:*

$$\mathbb{E}\left[g_{i,k}^t\right] = \nabla f_i(w_{i,k}^t), \quad \mathbb{E}\|g_{i,k}^t - \nabla f_i(w_{i,k}^t)\|^2 \le \sigma_l^2. \quad (5)$$

**Assumption 3** *For $\forall w \in \mathbb{R}^d$, the heterogeneous similarity is bounded on the gradient norm as:*

$$\mathbb{E}\|\nabla f_i(w)\|^2 \le G^2 + B^2 \mathbb{E}\|\nabla f(w)\|^2. \quad (6)$$

**Assumption 4** *For $\forall w_1, w_2 \in \mathbb{R}^d$, the global function $f$ satisfies $L_G$-Lipschitz if:*

$$\|f(w_1) - f(w_2)\| \le L_G\|w_1 - w_2\|. \quad (7)$$

**Assumption 5** *For $\forall w \in \mathbb{R}^d$, let $w^* \in \arg\min_w f(w)$, the function $f$ satisfies PŁ-condition if:*

$$2\mu\left(f(w) - f(w^*)\right) \le \|\nabla f(w)\|^2. \quad (8)$$

**Discussions.** Assumptions 1~3 are three general assumptions to analyze the non-convex objective in FL, which is widely used in the previous works [15, 18, 19, 30, 36, 44, 46, 48]. Assumption 4 is used to bound the uniform stability for the non-convex objective, which is used in [11, 51]. Different from the analysis in the margin-based generalization bound [27, 29, 31, 38] that focus on understanding how the designed objective affects the final generalization performance, our work focuses on understanding

how the generalization performance changes in the training process. We consider the entire training process and adopt uniform stability to measure the global generality in FL. For the general non-convex objective, one often uses the gradient norm $\mathbb{E}\|\nabla f(w)\|^2$ instead of bounding the loss difference $\mathbb{E}\left[f(w^T) - f(w^\star)\right]$ to measure the optimization convergence. To construct and analyze the excess risk, and further understand how the consistency affects the FL paradigm, we follow [51] to use Assumption 5 to bound the loss distance. Through this, we can establish a theoretical framework to jointly analyze the trade-off on the optimization and generalization in the FL paradigm.

## 4.3 Main Theorems

### 4.3.1 Optimization Error $\mathcal{E}_O$

**Theorem 1** *Under Assumptions 1∼3, let participation ratio is $N/C$ where $1 < N < C$, let the learning rate satisfy $\eta \le \min\left\{\frac{N}{2CKL}, \frac{1}{NKL}\right\}$ where $K \ge 2$, let the relaxation coefficient $\beta \le \frac{\sqrt{2}}{12}$, and after training $T$ rounds, the global model $w^t$ generated by FedInit satisfies:*

$$\frac{1}{T}\sum_{t=0}^{T-1}\mathbb{E}\|f(w^t)\|^2 \le \frac{2\left(f(w^0) - f(w^\star)\right)}{\lambda\eta KT} + \frac{\kappa_2\eta L}{\lambda N}\sigma_l^2 + \frac{3\kappa_1\eta KL}{\lambda N}G^2, \tag{9}$$

*where $\lambda \in (0,1)$, $\kappa_1 = \frac{1300\beta^2}{1-72\beta^2} + 17$, and $\kappa_2 = \frac{1020\beta^2}{1-72\beta^2} + 13$ are three constants. Further, by selecting the proper learning rate $\eta = \mathcal{O}(\sqrt{\frac{N}{KT}})$ and let $D = f(w^0) - f(w^\star)$ as the initialization bias, the global model $w^t$ satisfies:*

$$\frac{1}{T}\sum_{t=0}^{T-1}\mathbb{E}\|f(w^t)\|^2 \le \mathcal{O}\left(\frac{D + L\left(\sigma_l^2 + KG^2\right)}{\sqrt{NKT}}\right). \tag{10}$$

Theorem 1 provides the convergence rate of the *FedInit* method without the *PŁ-condition*, which could achieve the $\mathcal{O}(1/\sqrt{NKT})$ with the linear speedup of $N\times$. The dominant term of the training convergence rate is the heterogeneous bias $G$, which is $K\times$ larger than the initialization bias $D$ and stochastic bias $\sigma_l$. According to the formulation (10), by ignoring the initialization bias, the best local interval $K = \mathcal{O}(\sigma_l^2/G^2)$. This selection also implies that when $G$ increases, which means the local heterogeneity increases, the local interval $K$ is required to decrease appropriately to maintain the same efficiency. More importantly, though *FedInit* adopts a weighted bias on the initialization state at the beginning of each communication round, the divergence term $\mathbb{E}\|w_{i,K}^{t-1} - w^t\|^2$ does not affect the convergence bound whether $\beta$ is 0 or not. This indicates that the FL paradigm allows a divergence of local clients from the optimization perspective. Proof details are stated in Appendix A.2.3.

**Theorem 2** *Under Assumptions 1∼3 and 5, let participation ratio is $N/C$ where $1 < N < C$, let the learning rate satisfy $\eta \le \min\left\{\frac{N}{2CKL}, \frac{1}{NKL}, \frac{1}{\lambda\mu K}\right\}$ where $K \ge 2$, let the relaxation coefficient $\beta \le \frac{\sqrt{2}}{12}$, and after training $T$ rounds, the global model $w^t$ generated by FedInit satisfies:*

$$\mathbb{E}[f(w^T) - f(w^\star)] \le e^{-\lambda\mu\eta KT}\mathbb{E}[f(w^0) - f(w^\star)] + \frac{3\kappa_1\eta KL}{2N\lambda\mu}G^2 + \frac{\kappa_2\eta L}{2N\lambda\mu}\sigma_l^2, \tag{11}$$

*where $\lambda, \kappa_1, \kappa_2$ is defined in Theorem 1. Further, by selecting the proper learning rate $\eta = \mathcal{O}(\frac{\log(\lambda\mu NKT)}{\lambda\mu KT})$ and let $D = f(w^0) - f(w^\star)$ as the initialization bias, the global model $w^t$ satisfies:*

$$\mathbb{E}[f(w^T) - f(w^\star)] = \widetilde{\mathcal{O}}\left(\frac{D + L(\sigma_l^2 + KG^2)}{NKT}\right). \tag{12}$$

To bound the $\mathcal{E}_O$ term, we adopt Assumption 5 and prove that *FedInit* method could achieve the $\mathcal{O}(1/NKT)$ rate where we omit the $\mathcal{O}(\log(NKT))$ term. It maintains the properties stated in the Theorem 1. Detailed proofs of the convergence bound are stated in Appendix A.2.4.

### 4.3.2 Generalization Error $\mathcal{E}_G$

**Uniform Stability.** One powerful analysis of the generalization error is the uniform stability [11, 21, 50]. It says, for a general proposed method, its generalization error is always lower than the bound

of uniform stability. We assume that there is a new set $\widetilde{\mathcal{C}}$ where $\mathcal{C}$ and $\widetilde{\mathcal{C}}$ differ in at most one data sample on the $i^\star$-th client. Then we denote the $w^T$ and $\widetilde{w}^T$ as the generated model after training $T$ rounds on these two sets, respectively. Thus, we have the following lemma:

**Lemma 1** *(**Uniform Stability.** [11]) For the two models $w^T$ and $\widetilde{w}^T$ generated as introduced above, a general method satisfies $\epsilon$-uniformly stability if:*

$$\sup_{z_j \sim \{\mathcal{D}_i\}} \mathbb{E}[f(w^T; z_j) - f(\widetilde{w}^T; z_j)] \leq \epsilon. \tag{13}$$

*Moreover, if a general method satisfies $\epsilon$-uniformly stability, then its generalization error satisfies $\mathcal{E}_G \leq \sup_{z_j \sim \{\mathcal{D}_i\}} \mathbb{E}[f(w^T; z_j) - f(\widetilde{w}^T; z_j)] \leq \epsilon$ [50].*

**Theorem 3** *Under Assumptions 1, 2, 4, and 5, let all conditions above be satisfied, let learning rate $\eta = \mathcal{O}(\frac{1}{KT}) = \frac{c}{T}$ where $c = \frac{\mu_0}{K}$ is a constant, and let $|\mathcal{S}_i| = S$ as the number of the data samples, by randomly selecting the sample $z$, we can bound the uniform stability of our proposed FedInit as:*

$$\mathbb{E}\|f(w^{T+1}; z) - f(\widetilde{w}^{T+1}; z)\|$$
$$\leq \frac{1}{S-1}\left[\frac{2(L_G^2 + SL_G\sigma_l)(UTK)^{cL}}{L}\right]^{\frac{1}{1+cL}} + (1+\beta)^{\frac{1}{\beta cL}}\left[\frac{ULTK}{2(L_G^2 + SL_G\sigma_l)}\right]^{\frac{cL}{1+cL}} \sum_{t=1}^{T}\frac{\sqrt{\Delta^t}}{T}, \tag{14}$$

*where $U$ is a constant and $\Delta^t = \frac{1}{C}\sum_{i\in\mathcal{C}} \mathbb{E}\|w_{i,K}^{t-1} - w^t\|^2$ is the divergence term at round $t$.*

For the generalization error, Theorem 3 indicates that $\mathcal{E}_G$ term contains two main parts. The first part comes from the stochastic gradients as the vanilla centralized training process [11], which is of the order $\mathcal{O}((TK)^{\frac{cL}{1+cL}}/S)$. The constant $c$ is of the order $\mathcal{O}(1/K)$ as $c = \frac{\mu_0}{K}$, thus we have $\frac{cL}{1+cL} = \frac{\mu_0 L}{K+\mu_0 L}$. If we assume the $\mu_0 L$ is generally small, we always expect to adopt a larger $K$ in the FL paradigm to reduce generalization error. For instance, if we select $K \to \infty$, then $\mathcal{O}((TK)^{\frac{cL}{1+cL}}/S) \to \mathcal{O}(T^{\frac{cL}{1+cL}}/S)$ which is a very strong upper bound of the generalization error. However, the selection of local interval $K$ must be restricted from the optimization conditions and we will discuss the details in Section 4.3.4. In addition, the second part in Theorem 3 comes from the divergence term, which is a unique factor in the FL paradigm. As we mentioned above, the divergence term measures the authentic client-drift in the training process. The divergence term is not affected by the number of samples $S$ and it is only related to the proposed method and the local heterogeneity of the dataset. Proof details are stated in Appendix A.3.

### 4.3.3 Divergence Term

In the former two parts, we provide the complete theorem to measure optimization error $\mathcal{E}_O$ and generalization error $\mathcal{E}_G$. And we notice that, in the FL paradigm, the divergence term mainly affects the generalization ability of the model instead of the optimization convergence. In this part, we focus on the analysis of the divergence term of our proposed *FedInit* method. Due to the relaxed initialization at the beginning of each communication round, according to the Algorithm 1, we have $w_{i,K}^t = w^t + \beta(w^t - w_{i,K}^{t-1}) - \eta\sum_{k=0}^{K-1} g_{i,k}^t$. Thus, we have the following recursive relationship:

$$\underbrace{w^{t+1} - w_{i,K}^t}_{\text{local divergence at } t+1} = \beta\underbrace{(w_{i,K}^{t-1} - w^t)}_{\text{local divergence at } t} + \underbrace{(w^{t+1} - w^t)}_{\text{global update}} + \underbrace{\sum_{k=0}^{K-1}\eta g_{i,k}^t}_{\text{local updates}}. \tag{15}$$

According to the formulation (15), we can bound the divergence $\Delta^t$ via the following two theorems.

**Theorem 4** *Under Assumptions 1~3, we can bound the divergence term as follows. Let the learning rate satisfy $\eta \leq \min\left\{\frac{N}{2CKL}, \frac{1}{NKL}, \frac{\sqrt{N}}{\sqrt{C}KL}\right\}$ where $K \geq 2$, and after training $T$ rounds, let $0 < \beta < \frac{\sqrt{6}}{24}$, the divergence term $\{\Delta^t\}$ generated by FedInit satisfies:*

$$\frac{1}{T}\sum_{t=0}^{T-1}\Delta^t = \mathcal{O}\left(\frac{N(\sigma_l^2 + KG^2)}{T} + \frac{\sqrt{NK}B^2[D + L(\sigma_l^2 + KG^2)]}{T^{\frac{3}{2}}}\right). \tag{16}$$

Theorem 4 points out the convergence order of the divergence $\Delta^t$ generated by *FedInit* during the training process. This bound matches the conclusion in Theorem 1 with the same learning rate. The

dominant term achieves the $\mathcal{O}(NK/T)$ rate on the heterogeneity bias $G$. It could be seen that the number of selected clients $N$ will inhibit its convergence and the local consistency linearly increases with $N$. Different from the selection in Theorem 1, local interval $K$ is expected as small enough to maintain the high consistency. Also, the initialization bias $D$ is no longer dominant in consistency. We omit the constant weight $\frac{1}{1-96\beta^2}$ in this upper bound. Proof details are stated in Appendix A.2.5.

**Theorem 5** *Under Assumptions 1~3 and 5, we can bound the divergence term as follows. Let the learning rate satisfy $\eta \leq \min\left\{\frac{N}{2CKL}, \frac{1}{NKL}, \frac{1}{\lambda\mu K}\right\}$ where $K \geq 2$, and after training $T$ rounds, let $0 < \beta < \frac{\sqrt{6}}{24}$, the divergence term $\Delta^T$ generated by FedInit satisfies:*

$$\Delta^T = \widetilde{\mathcal{O}}\left(\frac{D + G^2}{T^2} + \frac{N\sigma_l^2 + KG^2}{NKT^2} + \frac{1}{NKT^3}\right). \tag{17}$$

Theorem 5 indicates the convergence of the divergence $\Delta$ under the *PŁ-condition* which matches the conclusion in Theorem 2 with the same learning rate selection. Assumption 5 establishes a relationship between the gradient norm and the loss difference on the non-convex function $f$. Different from the Theorem 4, the initialization bias $D$ and the heterogeneous bias $G$ are the dominant terms. Under Assumption 5, the *FedInit* supports a larger local interval $K$ in the training process. This conclusion also matches the selection of $K$ in Theorem 2. When the model converges, *FedInit* guarantees the local models towards the global optimum under at least $\mathcal{O}(1/T^2)$ rate. Similarly, we omit the constant weight $\frac{1}{1-96\beta^2}$ and we will discuss the $\beta$ in Section 4.3.4. Proof details are stated in Appendix A.2.6.

#### 4.3.4 Excess Risk

In this part, we analyze the excess risk $\mathcal{E}_E$ of *FedInit* method. According to the theorems above,

**Theorem 6** *Under Assumption 1~5, let the participation ratio is $N/C$ where $1 < N < C$, let the learning rate satisfies $\eta \leq \min\{\frac{N}{2CKL}, \frac{1}{NKL}, \frac{1}{\lambda\mu K},\}$ where $K \geq 2$, let the relaxed coefficient $0 \leq \beta < \frac{\sqrt{6}}{24}$, and let $|\mathcal{S}_i| = S$. By selecting the learning rate $\eta = \mathcal{O}(\frac{\log(\lambda\mu NKT)}{\lambda\mu KT}) \leq \frac{c}{t}$, after training $T$ communication rounds, the excess risk of the FedInit method achieves:*

$$\mathcal{E}_E \leq \underbrace{\widetilde{\mathcal{O}}\left(\frac{D + L(\sigma_l^2 + KG^2)}{NKT}\right)}_{\text{optimization bias}} + \underbrace{\mathcal{O}\left(\frac{1}{S}\left[\sigma_l(TK)^{cL}\right]^{\frac{1}{1+cL}}\right)}_{\text{stability bias}} + \underbrace{\widetilde{\mathcal{O}}\left(\frac{\sqrt{D + G^2}K^{\frac{cL}{1+cL}}}{T^{\frac{1}{1+cL}}}\right)}_{\text{divergence bias}}. \tag{18}$$

According to the Theorems 2, 3, and 5, we combine their dominant terms to upper bound the excess risk of *FedInit* method. The first term comes from the optimization error, the second term comes from the stability bias, and the third term comes from the divergence bias. From the perspective of excess risk, the main restriction in the FL paradigm is the divergence term with the bound of $\widetilde{\mathcal{O}}(\frac{1}{T^{\frac{1}{1+cL}}})$.

The second term of excess risk matches the conclusion in SGD [11, 51] which relies on the number $S$. Our analysis of the excess risk reveals two important corollaries in FL:

- From the perspective of optimization, the FL paradigm is insensitive to local consistency in the training process (Theorems 1&2).
- From the perspective of generalization, the local consistency level significantly affects the performance in the FL paradigm (Theorem 6).

Then we discuss the best selection of the local interval $K$ and relaxed coefficient $\beta$.

**Selection of K.** In the first term, to minimize the optimization error, the local interval $K$ is required to be large enough. In the second term, since $\frac{cL}{1+cL} \leq 1$, the upper bound expects a small local interval $K$. In the third term, since $\frac{1}{1+cL} = \frac{K}{K+\mu_0 L} < 1$, it expects a large $K$ to guarantee the order of $T$ to approach $\mathcal{O}(1/T)$, where the divergence bias could maintain a high-level consistency. Therefore, there is a specific optimal constant selection for $K > 1$ to minimize the excess risk.

**Selection of $\beta$.** As the dominant term, the coefficient of the divergence bias also plays a key role in the error bound. In Theorem 5, the constant weight we omit for the divergence term $\Delta^T$ is $\frac{1}{1-96\beta^2}$. Thus the coefficient of $\sqrt{\Delta^T}$ is $\frac{1}{\sqrt{1-96\beta^2}}$. Combined with Theorem 3, we have the coefficient for the

Table 1: Test accuracy (%) on the CIFAR-10/100 dataset. We test two participation ratios on each dataset. Under each setup, we test two Dirichlet splittings, and each result test for 3 times. This table reports results on ResNet-18-GN (upper part) and VGG-11 (lower part) respectively.

| Method | CIFAR-10 | | | | CIFAR-100 | | | |
| --- | --- | --- | --- | --- | --- | --- | --- | --- |
| | 10%-100 clients | | 5%-200 clients | | 10%-100 clients | | 5%-200 clients | |
| | Dir-0.6 | Dir-0.1 | Dir-0.6 | Dir-0.1 | Dir-0.6 | Dir-0.1 | Dir-0.6 | Dir-0.1 |
| FedAvg | $78.77_{\pm.11}$ | $72.53_{\pm.17}$ | $74.81_{\pm.18}$ | $70.65_{\pm.21}$ | $46.35_{\pm.15}$ | $42.62_{\pm.22}$ | $44.70_{\pm.22}$ | $40.41_{\pm.33}$ |
| FedAdam | $76.52_{\pm.14}$ | $70.44_{\pm.22}$ | $73.28_{\pm.18}$ | $68.87_{\pm.26}$ | $48.35_{\pm.17}$ | $40.77_{\pm.31}$ | $44.33_{\pm.26}$ | $38.04_{\pm.25}$ |
| FedSAM | $79.23_{\pm.22}$ | $72.89_{\pm.23}$ | $75.45_{\pm.19}$ | $71.23_{\pm.26}$ | $47.51_{\pm.26}$ | $43.43_{\pm.12}$ | $45.98_{\pm.27}$ | $40.22_{\pm.27}$ |
| SCAFFOLD | $81.37_{\pm.17}$ | $75.06_{\pm.16}$ | $78.17_{\pm.28}$ | $74.24_{\pm.22}$ | $51.98_{\pm.23}$ | $\mathbf{44.41}_{\pm.15}$ | $50.70_{\pm.29}$ | $41.83_{\pm.29}$ |
| FedDyn | $82.43_{\pm.16}$ | $75.08_{\pm.19}$ | $79.96_{\pm.13}$ | $74.15_{\pm.34}$ | $50.82_{\pm.19}$ | $42.50_{\pm.28}$ | $47.32_{\pm.21}$ | $41.74_{\pm.21}$ |
| FedCM | $81.67_{\pm.17}$ | $73.93_{\pm.26}$ | $79.49_{\pm.17}$ | $73.12_{\pm.18}$ | $51.56_{\pm.20}$ | $43.03_{\pm.26}$ | $50.93_{\pm.19}$ | $42.33_{\pm.19}$ |
| **FedInit** | $\mathbf{83.11}_{\pm.29}$ | $\mathbf{75.95}_{\pm.19}$ | $\mathbf{80.58}_{\pm.20}$ | $\mathbf{74.92}_{\pm.17}$ | $\mathbf{52.21}_{\pm.09}$ | $44.22_{\pm.21}$ | $\mathbf{51.16}_{\pm.18}$ | $\mathbf{43.77}_{\pm.36}$ |
| FedAvg | $85.28_{\pm.12}$ | $78.02_{\pm.22}$ | $81.23_{\pm.14}$ | $74.89_{\pm.25}$ | $53.46_{\pm.25}$ | $50.53_{\pm.20}$ | $47.55_{\pm.13}$ | $45.05_{\pm.33}$ |
| FedAdam | $86.44_{\pm.13}$ | $77.55_{\pm.28}$ | $81.05_{\pm.23}$ | $74.04_{\pm.17}$ | $55.56_{\pm.29}$ | $53.41_{\pm.18}$ | $51.33_{\pm.25}$ | $47.26_{\pm.21}$ |
| FedSAM | $86.37_{\pm.22}$ | $79.10_{\pm.07}$ | $81.76_{\pm.26}$ | $75.22_{\pm.13}$ | $54.85_{\pm.31}$ | $51.88_{\pm.27}$ | $48.65_{\pm.21}$ | $46.58_{\pm.28}$ |
| SCAFFOLD | $87.73_{\pm.17}$ | $81.98_{\pm.19}$ | $84.81_{\pm.15}$ | $79.04_{\pm.16}$ | $\mathbf{59.45}_{\pm.17}$ | $56.67_{\pm.24}$ | $53.73_{\pm.32}$ | $50.08_{\pm.19}$ |
| FedDyn | $87.35_{\pm.19}$ | $82.70_{\pm.24}$ | $84.84_{\pm.19}$ | $\mathbf{80.01}_{\pm.22}$ | $56.13_{\pm.18}$ | $53.97_{\pm.11}$ | $51.74_{\pm.18}$ | $48.16_{\pm.17}$ |
| FedCM | $86.80_{\pm.33}$ | $79.85_{\pm.29}$ | $83.23_{\pm.31}$ | $76.42_{\pm.36}$ | $53.88_{\pm.22}$ | $50.73_{\pm.35}$ | $47.83_{\pm.19}$ | $46.33_{\pm.25}$ |
| **FedInit** | $\mathbf{88.47}_{\pm.22}$ | $\mathbf{83.51}_{\pm.13}$ | $\mathbf{85.36}_{\pm.19}$ | $79.73_{\pm.14}$ | $58.84_{\pm.11}$ | $\mathbf{57.22}_{\pm.21}$ | $\mathbf{54.12}_{\pm.08}$ | $\mathbf{50.27}_{\pm.29}$ |

divergence term in formulation (18) is $\frac{(1+\beta)^{\frac{1}{\beta cL}}}{\sqrt{1-96\beta^2}}$. Therefore, to minimize this term, there is a specific optimal constant selection for $0 < \beta < \frac{\sqrt{6}}{24}$. We validate their selections in Section 5.2.

## 5  Experiments

In this part, we introduce our empirical studies. Due to the page limitations, the details of the dataset, hyperparameters selection, implementation, and some extra ablation studies are stated in Appendix B.

**Benchmarks.** Our selected benchmarks in this paper are stated as follows. *FedAvg* [26] proposes the general FL paradigm. *FedAdam* [30] studies the efficiency of adaptive optimizer in FL. *SCAFFOLD* [19], *FedDyn* [1], and *FedCM* [46] learn the "client-drift" problem and adopt the variance reduction technique, ADMM, and client-level momentum respectively in FL to alleviate its negative impact. *FedSAM* [29] uses the local SAM objective instead of the vanilla empirical risk objective to search for a smooth loss landscape, which focuses on the generalization performance.

**Setups.** Here we briefly introduce the setups in our experiments. We test our proposed *FedInit* on the CIFAR-10 /100 dataset [20]. To generate local heterogeneity, we follow Hsu et al. [14] to split the local clients through the Dirichlet sampling via a coefficient $D_r$ to control the heterogeneous level and follow Sun et al. [38] to adopt the sampling with replacement to enhance the heterogeneity level. We test on the ResNet-18-GN [12, 13] and VGG-11 [35] to validate its efficiency. Actually, when the heterogeneity is strong, the performance of personalized initialization will be better. To better demonstrate the performance of our proposed method, we add additional noises to the dataset. Specifically, we first introduce the *client-based biases*. Among clients, we assume that the data samples are obtained differently. Because the local dataset is private and its construction is unknown, i.e., they are collected from different machines or cameras. Therefore, we change the strength of the $RGB$ channels with a random Gaussian noise for different clients. The second noise is the *category-based biases*. We assume that samples for each category also contain heterogeneity. In our experiments, we add different brightness perturbations to the samples in each category by a random Gaussian noise. Based on these two noises, the heterogeneity of the local dataset is significantly enlarged. In this more realistic dataset, we can clearly observe the performance of each algorithm.

For each benchmark in our experiments, we adopt two coefficients $D_r = 0.1$ and $0.6$ for each dataset to generate different heterogeneity. We generally select the local learning rate $\eta = 0.1$ and global learning rate $\eta = 1$ on all setups except for *FedAdam* we use $0.1$. The learning rate decay is set as multiplying $0.998$ per round except for *FedDyn* we use $0.999$. We train 500 rounds on CIFAR-10 and 800 rounds on CIFAR-100 to achieve stable test accuracy. The participation ratios are selected as 10% and 5% respectively of total 100 and 200 clients. More details are stated in Appendix B.1.

## 5.1 Experiment results

In this part, we mainly introduce the experiment results compared with the other benchmarks.

In Table 1, our proposed *FedInit* method performs well than the other benchmarks with good stability across different experimental setups. On the results of ResNet-18-GN on CIFAR-10, it achieves about $3.42\%$ improvement than the vanilla *FedAvg* on the high heterogeneous splitting with $D_r = 0.1$. When the participation ratio decreases to $5\%$, the accuracy drops only about $0.1\%$ while *FedAvg* drops almost $1.88\%$. Similar results on CIFAR-100, when the ratio de-

Table 2: We incorporate the relaxed initialization (RI) into the benchmarks to test improvements on ResNet-18-GN on CIFAR-10 with the same hyperparameters and specific relaxed coefficient $\beta$.

| Method | 10%-100 clients | | | | 5%-200 clients | | | |
| | Dir-0.6 | | Dir-0.1 | | Dir-0.6 | | Dir-0.1 | |
| | - | +RI | - | +RI | - | +RI | - | +RI |
|---|---|---|---|---|---|---|---|---|
| FedAvg | 78.77 | 83.11 | 72.53 | 75.95 | 74.81 | 80.58 | 70.65 | 74.92 |
| FedAdam | 76.52 | 78.33 | 70.44 | 72.55 | 73.28 | 78.33 | 68.87 | 71.34 |
| FedSAM | 79.23 | **83.36** | 72.89 | 76.34 | 75.45 | 80.66 | 71.23 | 75.08 |
| SCAFFOLD | 81.37 | 83.27 | 75.06 | **77.30** | 78.17 | **81.02** | 74.24 | **76.22** |
| FedDyn | 82.43 | 81.91 | 75.08 | 75.11 | 79.96 | 79.88 | 74.15 | 74.34 |
| FedCM | 81.67 | 81.77 | 73.93 | 73.71 | 79.49 | 79.72 | 73.12 | 72.98 |

creases, *FedInit* still achieves $43.77\%$ while the second best method *SCAFFOLD* drops about $3.21\%$. This indicates the proposed *FedInit* holds good stability on the varies of the participation. In addition, in Table 2, we incorporate the relaxed initialization (RI) into the other benchmarks to test its benefit. "-" means the vanilla benchmarks, and "+RI" means adopting the relaxed initialization. It shows that the relaxed initialization holds the promising potential to further enhance the performance. Actually, *FedInit* could be considered as (RI + *FedAvg*), whose improvement achieves about over $3\%$ on each setup. Table 1 shows the poor performance of the vanilla *FedAvg*. Nevertheless, when adopting the RI, *FedInit* remains above most benchmarks on several setups. When the RI is incorporated into other benchmarks, it helps them to achieve higher performance without additional communication costs.

## 5.2 Ablation

In this part, we mainly introduce the ablation results of different hyperparameters.

**Hyperparameters Sensitivity.** The excess risk and test error of *FedInit* indicate there exists best selections for local interval $K$ and relaxed coefficient $\beta$, respectively. In this part, we test a series of selections to validate our conclusions. To be aligned with previous studies, we denote $K$ as training epochs. In Figure 1 (a), we can see that the selection range of the beta is very small while it has great potential to improve performance. When it is larger than the threshold, the training process will diverge quickly. As local interval $K$ increases, test accuracy rises first

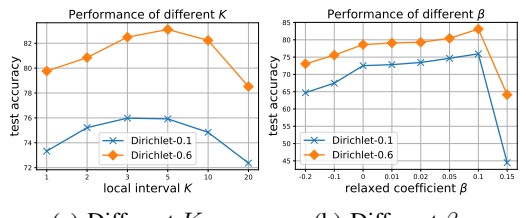

(a) Different $K$.      (b) Different $\beta$.

Figure 1: THyperparameters sensitivity studies of local intervals $K$ and relaxed coefficient $\beta$ of the *FedInit* method on CIFAR-10. To fairly compare their efficiency, we fix the total communication rounds $T = 500$.

and then decreases. Our analysis provides a clear explanation of the phenomenon. The optimization error decreases as $K$ increases when it is small. When $K$ exceeds the threshold, the divergence term in generalization cannot be ignored. Therefore, the test accuracy will be significantly affected.

**Consistency.** In this part, we test the relationship between the test accuracy and divergence term $\Delta^T$ under different $\beta$ selections. As introduced in Algorithm 1 Line.6, negative $\beta$ means to adopt the relaxed initialization which is close to the latest local model. *FedInit* degrades to *FedAvg* when $\beta = 0$. Table 3 validates that RI is required to

Table 3: We test different selections of the relaxed coefficient $\beta$ of the *FedInit* method on CIFAR-10 10%-100 Dir-0.1 splitting to validate the relationship between test error and consistency after 500 rounds. We fix other hyperparameters as the same selection above for a fair comparison.

| $\beta$ | -0.2 | -0.1 | 0 | 0.01 | 0.02 | 0.05 | 0.1 | 0.15 |
|---|---|---|---|---|---|---|---|---|
| Accuracy (%) | 64.70 | 67.47 | 72.53 | 72.82 | 73.45 | 74.65 | **75.95** | 44.47 |
| $\Delta^T$ | 0.873 | 0.815 | 0.855 | 0.875 | 0.850 | 0.823 | **0.760** | $\infty$ |

be far away from the local model (a positive $\beta$). When $\beta$ is small, the correction is limited. The local divergence term is difficult to be diminished efficiently. While it becomes too large, the local training begins from a bad initialization, which can not receive enough guidance of global information from the global models. Furthermore, as shown in Table 3, if the initialization is too far from the local model, the quality of the initialization state will not be effectively guaranteed.

### 5.3 Discussions of Relaxed Initialization

In this part, we mainly discuss the improvements of the proposed relaxed initialization.

In vanilla classical *FedAvg* and the most advanced methods, at the beginning of each communication round, we are always caught in a misunderstanding of the high consistency. Because the target of FL is a globally consistent solution, it is always an involuntary aggregation in the algorithm to ensure consistency. We prove that this does contribute to the efficiency of the optimization process, but it is not the best selection for generalization. To better improve the generalization, we prove that a relaxed initialization state will contribute more. We compare their difference in Figure 5.3.

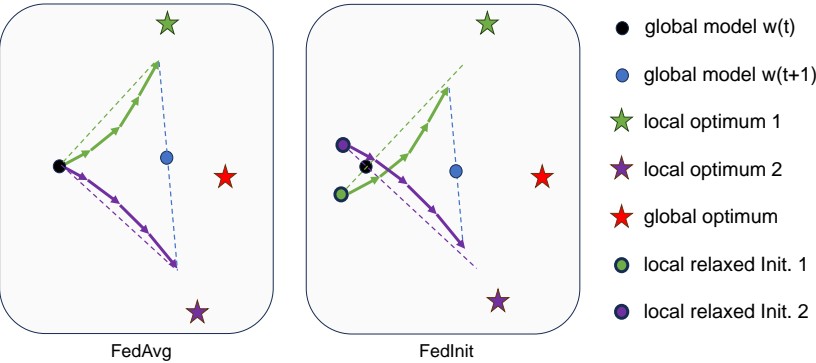

Figure 2: Schematics of the classical *FedAvg* and our proposed *FedInit*.

As shown in the above figure, we can clearly see why *FedInit* contributes more to the consistency. When we move a little in the opposite direction of the last local optimization state, we will move further away from local optimal solutions in the current communication round. The working mode of RI is similar to the idea of "lookahead". Differently, (1) "lookahead" only works at the end of each stage; (2) "lookahead" only works for the global models on the global server. However, RI helps each local client to backtrack a small distance at the beginning of each stage. Therefore, after the local training in the next stage, the trained local models will get closer to each other than before.

## 6  Conclusion

In this work, we propose an efficient and novel FL method, dubbed *FedInit*, which adopts the stage-wise personalized relaxed initialization to enhance the local consistency level. Furthermore, to clearly understand the essential impact of consistency in FL, we introduce the excess risk analysis in FL and study the divergence term. Our proofs indicate that consistency dominates the test error and generalization error bound while optimization error is insensitive to it. Extensive experiments are conducted to validate the efficiency of relaxed initialization. As a practical and light plug-in, it could also be easily incorporated into other FL paradigms to improve their performance.

**Limitations & Broader Impact.** In this work, we analyze the excess risk for the *FedInit* method to understand how consistency works in FL. Actually, the relaxed initialization may also work for the personalized FL (pFL) paradigm. It is a future study to explore its properties in the pFL and decentralized FL, which may inspire us to design novel efficient algorithms in the FL community.

## Acknowledgements

Prof. Dacheng Tao is partially supported by Australian Research Council Project FL-170100117.

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

# A Proofs

In this section, we introduce our proofs of the main theorems in the main context. In the first part, we introduce some assumptions used in our proofs and point out their functions used for which part. In the second part, we prove the convergence rate and optimization error under the general assumptions. In the third part, we prove the uniform stability to measure the generalization error and analyze how each term affects the accuracy.

We suppose there are $C$ clients participating in the training process and each has a local heterogeneous dataset. In each round $t$, we randomly select $N$ clients to send the global model and they will train $K$ iterations to get $N$ local models. The local models will be aggregated on the global server as the next global model. After $T$ rounds, our method generates a global model as the final state. We denote the total client set as $\mathcal{C}$ and the selected client set as $\mathcal{N}$.

## A.1 Assumptions

In this part, we state assumptions in our proofs and discuss them. We will introduce each assumption and develop their corollaries.

**Assumption 6** *For $\forall w_1, w_2 \in \mathbb{R}^d$, the non-convex local function $f_i$ satisfies L-smooth if:*

$$\|\nabla f_i(w_1) - \nabla f_i(w_2)\| \leq L\|w_1 - w_2\|, \tag{19}$$

*where $L$ is a universal constant.*

**Assumption 7** *For $\forall w \in \mathbb{R}^d$, the stochastic gradient is bounded by its expectation and variance as:*

$$\begin{aligned} \mathbb{E}\left[g_{i,k}^t\right] &= \nabla f_i(w_{i,k}^t), \\ \mathbb{E}\|g_{i,k}^t - \nabla f_i(w_{i,k}^t)\|^2 &\leq \sigma_l^2, \end{aligned} \tag{20}$$

*where $\sigma_l > 0$ is a universal constant.*

**Assumption 8** *For $\forall w \in \mathbb{R}^d$, the heterogeneous similarity is bounded on the gradient norm as:*

$$\frac{1}{C} \sum_{i \in \mathcal{C}} \|\nabla f_i(w)\|^2 \leq G^2 + B^2 \|\nabla f(w)\|^2, \tag{21}$$

*where $G \geq 0$ and $B \geq 1$ are two universal constants.*

**Assumption 9** *For $\forall w_1, w_2 \in \mathbb{R}^d$, the global function $f$ satisfies $L_G$-Lipschitz if:*

$$\|f(w_1) - f(w_2)\| \leq L_G\|w_1 - w_2\|, \tag{22}$$

*where $L_G$ is a universal constant.*

**Assumption 10** *For $\forall w \in \mathbb{R}^d$, let $w^\star \in \arg\min_w f(w)$, the global function satisfies PŁ-condition if:*

$$2\mu\left(f(w) - f(w^\star)\right) \leq \|\nabla f(w)\|^2, \tag{23}$$

*where $\mu$ is a universal positive constant.*

**Discussion.** Assumption 6~8 are three general assumptions to analyze the non-convex objective in FL, which is widely used in the previous works [15, 18, 19, 30, 36, 44, 46, 48]. Assumption 9 is used to bound the uniform stability for the non-convex objective, which is used in [11, 51]. Different from the analysis in the margin-based generalization bound [27, 29, 31, 38] that focus on understanding how the designed objective affects the final generalization performance, our work focuses on understanding how the generalization performance changes in the training process. We consider the entire training process and adopt uniform stability to measure the global generality in FL and theoretically study the importance of consistency to FL. For the general non-convex objective, one often uses the gradient norm $\mathbb{E}\|\nabla f(w)\|^2$ instead of the loss difference $\mathbb{E}\left[f(w^\star) - f(w)\right]$ to measure the training error. To construct and analyze the *excess risk* to further understand how the consistency affects the FL paradigm, we follow [51] to use Assumption 10 to bound the loss distance. Through this, we can establish a theoretical framework to jointly analyze the trade-off on the optimization and generalization in the FL paradigm.

## A.2 Proofs for the Optimization Error

In this part, we prove the training error for our proposed method. We assume the objective function $f(w) = \frac{1}{C} \sum_{i \in \mathcal{C}} f_i(w)$ is $L$-smooth w.r.t $w$. Then we could upper bound the training error in the FL. Some useful notations in the proof are introduced in the Table 4.

Then we introduce some important lemmas used in the proof.

Table 4: Some abbreviations of the used terms in the proof of bounded training error.

| Notation | Formulation | Description |
|---|---|---|
| $w_{i,k}^t$ | - | parameters at $k$-th iteration in round $t$ on client $i$ |
| $w^t$ | - | global parameters in round $t$ |
| $V_1^t$ | $\frac{1}{C}\sum_{i\in\mathcal{C}}\sum_{k=0}^{K-1}\mathbb{E}\|w_{i,k}^t - w^t\|^2$ | averaged norm of the local updates in round $t$ |
| $V_2^t$ | $\mathbb{E}\|w^{t+1} - w^t\|^2$ | norm of the global updates in round $t$ |
| $\Delta^t$ | $\frac{1}{C}\sum_{i\in\mathcal{C}}\mathbb{E}\|w_{i,K}^{t-1} - w^t\|^2$ | inconsistency/divergence term in round $t$ |
| $D$ | $f(w^0) - f(w^\star)$ | bias between the initialization state and optimal |

### A.2.1 Important Lemmas

**Lemma 2** (Bounded local updates) *We first bound the local training updates in the local training. Under the Assumptions stated, the averaged norm of the local updates of total $C$ clients could be bounded as:*

$$V_1^t \le 4K\beta^2\Delta^t + 3K^2\eta^2\left(\sigma_l^2 + 4KG^2\right) + 12K^3\eta^2 B^2\mathbb{E}\|\nabla f(w^t)\|^2. \tag{24}$$

**Proof** $V_1$ *measures the norm of the local offset during the local training stage. It could be bounded by two major steps. Firstly, we bound the separated term on the single client $i$ at iteration $k$ as:*

$$\mathbb{E}_t\|w^t - w_{i,k}^t\|^2$$
$$= \mathbb{E}_t\|w^t - w_{i,k-1}^t + \eta\left(g_{i,k-1}^t - \nabla f_i(w_{i,k-1}^t) + \nabla f_i(w_{i,k-1}^t) - \nabla f_i(w^t) + \nabla f_i(w^t)\right)\|^2$$
$$\le \left(1 + \frac{1}{2K-1}\right)\mathbb{E}_t\|w^t - w_{i,k-1}^t + \eta\left(g_{i,k-1}^t - \nabla f_i(w_{i,k-1}^t)\right)\|^2$$
$$\quad + 2K\eta^2\mathbb{E}_t\|\nabla f_i(w_{i,k-1}^t) - \nabla f_i(w^t) + \nabla f_i(w^t)\|^2$$
$$\le \left(1 + \frac{1}{2K-1}\right)\mathbb{E}_t\|w^t - w_{i,k-1}^t\|^2 + \eta^2\mathbb{E}_t\|g_{i,k-1}^t - \nabla f_i(w_{i,k-1}^t)\|^2$$
$$\quad + 4K\eta^2\mathbb{E}_t\|\nabla f_i(w_{i,k-1}^t) - \nabla f_i(w^t)\|^2 + 4K\eta^2\|\nabla f_i(w^t)\|^2$$
$$\le \left(1 + \frac{1}{2K-1} + 4\eta^2 KL^2\right)\mathbb{E}_t\|w^t - w_{i,k-1}^t\|^2 + \eta^2\sigma_l^2 + 4K\eta^2\|\nabla f_i(w^t)\|^2$$
$$\le \left(1 + \frac{1}{K-1}\right)\mathbb{E}_t\|w^t - w_{i,k-1}^t\|^2 + \eta^2\sigma_l^2 + 4K\eta^2\|\nabla f_i(w^t)\|^2,$$

*where the learning rate is required $\eta \le \frac{\sqrt{2}}{4(K-1)L}$ for $K \ge 2$.*

*Computing the average of the separated term on client $i$, we have:*

$$\frac{1}{C}\sum_{i\in\mathcal{C}}\mathbb{E}_t\|w^t - w_{i,k}^t\|^2$$
$$\le \left(1 + \frac{1}{K-1}\right)\frac{1}{C}\sum_{i\in\mathcal{C}}\mathbb{E}_t\|w^t - w_{i,k-1}^t\|^2 + \eta^2\sigma_l^2 + 4K\eta^2\frac{1}{C}\sum_{i\in\mathcal{C}}\|\nabla f_i(w^t)\|^2$$
$$\le \left(1 + \frac{1}{K-1}\right)\frac{1}{C}\sum_{i\in\mathcal{C}}\mathbb{E}_t\|w^t - w_{i,k-1}^t\|^2 + \eta^2\sigma_l^2 + 4K\eta^2 G^2 + 4K\eta^2 B^2\|\nabla f(w^t)\|^2.$$

*Unrolling the aggregated term on iteration $k \le K$. When local interval $K \ge 2$, $\left(1 + \frac{1}{K-1}\right)^k \le \left(1 + \frac{1}{K-1}\right)^K \le 4$. Then we have:*

$$\frac{1}{C}\sum_{i\in\mathcal{C}}\mathbb{E}_t\|w^t - w_{i,k}^t\|^2$$
$$\le \sum_{\tau=0}^{k-1}\left(1 + \frac{1}{K-1}\right)^\tau\left(\eta^2\sigma_l^2 + 4K\eta^2 G^2 + 4K\eta^2 B^2\|\nabla f(w^t)\|^2\right)$$
$$\quad + \left(1 + \frac{1}{K-1}\right)^k\frac{1}{C}\sum_{i\in\mathcal{C}}\|w^t - w_{i,0}^t\|^2$$
$$\le 3(K-1)\left(\eta^2\sigma_l^2 + 4K\eta^2 G^2 + 4K\eta^2 B^2\|\nabla f(w^t)\|^2\right) + 4\beta^2\frac{1}{C}\sum_{i\in\mathcal{C}}\mathbb{E}_t\|w^t - w_{i,K}^{t-1}\|^2$$

$$\leq 3K\eta^2 \left(\sigma_l^2 + 4KG^2\right) + 12K^2\eta^2 B^2 \|\nabla f(w^t)\|^2 + 4\beta^2 \Delta^t.$$

*Summing the iteration on $k = 0, 1, \cdots, K-1$,*

$$\frac{1}{C}\sum_{i\in\mathcal{C}}\sum_{k=0}^{K-1}\mathbb{E}_t\|w^t - w_{i,k}^t\|^2 \leq 4K\beta^2\Delta^t + 3K^2\eta^2\sigma_l^2 + 12K^3\eta^2 G^2 + 12K^3\eta^2 B^2\|\nabla f(w^t)\|^2.$$

*This completes the proof.*

**Lemma 3** (Bounded global updates) *The norm of the global update could be bounded by uniformly sampling. Under assumptions stated above, let $\eta \leq \frac{1}{KL}$, the norm of the global update of selected $N$ clients could be bounded as:*

$$\begin{aligned}
V_2^t \leq{}& \frac{15\beta^2}{N}\Delta^t + \frac{10\eta^2 K}{N}\sigma_l^2 + \frac{39\eta^2 K^2}{N}G^2 \\
&+ \frac{39\eta^2 K^2 B^2}{N}\mathbb{E}\|\nabla f(w^t)\|^2 + \frac{\eta^2}{CN}\mathbb{E}\|\sum_{i\in\mathcal{C}}\sum_{k=0}^{K-1}\nabla f_i(w_{i,k}^t)\|^2.
\end{aligned}$$
(25)

**Proof** $V_2$ *measures the variance of the global offset after each communication round. We define an indicator function $\mathbb{I}_{event} = 1$ if the event happens. Then, to bound it, we firstly split the expectation term:*

$$\begin{aligned}
&\mathbb{E}\|w^{t+1} - w^t\|^2 \\
={}& \mathbb{E}\|\frac{1}{N}\sum_{i\in\mathcal{N}}w_{i,K}^t - w^t\|^2 \\
={}& \frac{1}{N^2}\mathbb{E}\|\sum_{i\in\mathcal{N}}(w_{i,K}^t - w^t)\|^2 \\
={}& \frac{1}{N^2}\mathbb{E}\|\sum_{i\in\mathcal{C}}(w_{i,K}^t - w^t)\mathbb{I}_{i\in\mathcal{N}}\|^2 \\
={}& \frac{1}{N^2}\mathbb{E}\|\sum_{i\in\mathcal{C}}\mathbb{I}_{i\in\mathcal{N}}\left[\sum_{k=0}^{K-1}\eta g_{i,k}^t + \beta(w^t - w_{i,K}^{t-1})\right]\|^2 \\
={}& \frac{\eta^2}{NC}\sum_{i\in\mathcal{C}}\sum_{k=0}^{K-1}\mathbb{E}\|g_{i,k}^t - \nabla f_i(w_{i,k}^t)\|^2 + \frac{1}{N^2}\mathbb{E}\|\sum_{i\in\mathcal{C}}\mathbb{I}_{i\in\mathcal{N}}\left[\sum_{k=0}^{K-1}\eta\nabla f_i(w_{i,k}^t) + \beta(w^t - w_{i,K}^{t-1})\right]\|^2 \\
\leq{}& \frac{\eta^2 K\sigma_l^2}{N} + \frac{1}{N^2}\mathbb{E}\|\sum_{i\in\mathcal{C}}\mathbb{I}_{i\in\mathcal{N}}\left[\sum_{k=0}^{K-1}\eta\nabla f_i(w_{i,k}^t) + \beta(w^t - w_{i,K}^{t-1})\right]\|^2.
\end{aligned}$$

*To bound the second term, we can adopt the following equation. For the vector $x_i \in \mathbb{R}^d$, we have:*

$$\begin{aligned}
\mathbb{E}\|\sum_{i\in\mathcal{C}}\mathbb{I}_{i\in\mathcal{N}}x_i\|^2 ={}& \mathbb{E}\langle\sum_{i\in\mathcal{C}}\mathbb{I}_{i\in\mathcal{N}}x_i, \sum_{j\in\mathcal{C}}\mathbb{I}_{j\in\mathcal{N}}x_j\rangle \\
={}& \sum_{(i\neq j)\in\mathcal{C}}\mathbb{E}\langle\mathbb{I}_{i\in\mathcal{N}}x_i, \mathbb{I}_{j\in\mathcal{N}}x_j\rangle + \sum_{(i=j)\in\mathcal{C}}\mathbb{E}\langle\mathbb{I}_{i\in\mathcal{N}}x_i, \mathbb{I}_{j\in\mathcal{N}}x_j\rangle \\
={}& \sum_{(i\neq j)\in\mathcal{C}}\mathbb{E}\langle\mathbb{I}_{i\in\mathcal{N}}x_i, \mathbb{I}_{j\in\mathcal{N}}x_j\rangle + \sum_{(i=j)\in\mathcal{C}}\mathbb{E}\langle\mathbb{I}_{i\in\mathcal{N}}x_i, \mathbb{I}_{j\in\mathcal{N}}x_j\rangle \\
={}& \frac{N(N-1)}{C(C-1)}\sum_{(i\neq j)\in\mathcal{C}}\mathbb{E}\langle x_i, x_j\rangle + \frac{N}{C}\sum_{(i=j)\in\mathcal{C}}\mathbb{E}\langle x_i, x_j\rangle \\
={}& \frac{N(N-1)}{C(C-1)}\sum_{i,j\in\mathcal{C}}\mathbb{E}\langle x_i, x_j\rangle + \frac{N(C-N)}{C(C-1)}\sum_{(i=j)\in\mathcal{C}}\mathbb{E}\langle x_i, x_j\rangle \\
={}& \frac{N(N-1)}{C(C-1)}\mathbb{E}\|\sum_{i\in\mathcal{C}}x_i\|^2 + \frac{N(C-N)}{C(C-1)}\sum_{i\in\mathcal{C}}\mathbb{E}\|x_i\|^2.
\end{aligned}$$

*We firstly bound the first term in the above equation. Taking $x_i = \sum_{k=0}^{K-1}\eta\nabla f_i(w_{i,k}^t) + \beta(w^t - w_{i,K}^{t-1})$ into $\mathbb{E}\|\sum_{i\in\mathcal{C}}x_i\|^2$, we have:*

$$\mathbb{E}\|\sum_{i\in\mathcal{C}}\left[\sum_{k=0}^{K-1}\eta\nabla f_i(w_{i,k}^t) + \beta(w^t - w_{i,K}^{t-1})\right]\|^2 = \eta^2\mathbb{E}\|\sum_{i\in\mathcal{C}}\sum_{k=0}^{K-1}\nabla f_i(w_{i,k}^t)\|^2.$$

Then we bound the second term in above equation. Taking $x_i = \sum_{k=0}^{K-1} \eta \nabla f_i(w_{i,k}^t) + \beta(w^t - w_{i,K}^{t-1})$ into $\sum_{i\in\mathcal{C}} \mathbb{E}\|x_i\|^2$, we have:

$$\sum_{i\in\mathcal{C}} \mathbb{E}\| \sum_{k=0}^{K-1} \eta \nabla f_i(w_{i,k}^t) + \beta(w^t - w_{i,K}^{t-1})\|^2$$

$$= \sum_{i\in\mathcal{C}} \mathbb{E}\| \sum_{k=0}^{K-1} \left[ \eta \nabla f_i(w_{i,k}^t) + \frac{\beta}{K}(w^t - w_{i,K}^{t-1}) \right] \|^2$$

$$\leq K \sum_{i\in\mathcal{C}} \sum_{k=0}^{K-1} \mathbb{E}\| \eta \nabla f_i(w_{i,k}^t) + \frac{\beta}{K}(w^t - w_{i,K}^{t-1})\|^2$$

$$= K \sum_{i\in\mathcal{C}} \sum_{k=0}^{K-1} \mathbb{E}\| \eta \nabla f_i(w_{i,k}^t) - \eta \nabla f_i(w^t) + \eta \nabla f_i(w^t) + \frac{\beta}{K}(w^t - w_{i,K}^{t-1})\|^2$$

$$\leq 3\eta^2 K L^2 \underbrace{\sum_{i\in\mathcal{C}} \sum_{k=0}^{K-1} \mathbb{E}\|w_{i,k}^t - w^t\|^2}_{CV_1^t} + 3\eta^2 K^2 \sum_{i\in\mathcal{C}} \mathbb{E}\|\nabla f_i(w^t)\|^2 + 3\beta^2 \underbrace{\sum_{i\in\mathcal{C}} \mathbb{E}\|(w^t - w_{i,K}^{t-1})\|^2}_{C\Delta^t}$$

$$\leq 3C\eta^2 K L^2 V_1^t + 3C\beta^2 \Delta^t + 3C\eta^2 K^2 G^2 + 3C\eta^2 K^2 B^2 \mathbb{E}\|\nabla f(w^t)\|^2.$$

We bound all the components in $V_2^t$ term. Let $1 \leq N < C$, to generate the final bound, summarizing the inequalities all above and adopting the bounded $V_1^t$ in Lemma 2, then we have:

$$V_2^t \leq \frac{\eta^2 K \sigma_l^2}{N} + \frac{1}{N^2} \mathbb{E}\| \sum_{i\in\mathcal{C}} \mathbb{I}_{i\in\mathcal{N}} \left[ \sum_{k=0}^{K-1} \eta \nabla f_i(w_{i,k}^t) + \beta(w^t - w_{i,K}^{t-1}) \right] \|^2$$

$$\leq \frac{\eta^2 K \sigma_l^2}{N} + \frac{3(C-N)}{N(C-1)}(\eta^2 K L^2 V_1^t + \beta^2 \Delta^t + \eta^2 K^2 G^2 + \eta^2 K^2 B^2 \mathbb{E}\|\nabla f(w^t)\|^2)$$

$$+ \frac{(N-1)}{CN(C-1)} \eta^2 \mathbb{E}\| \sum_{i\in\mathcal{C}} \sum_{k=0}^{K-1} \nabla f_i(w_{i,k}^t)\|^2$$

$$\leq \frac{\eta^2 K \sigma_l^2}{N} + \frac{3}{N} \left( \beta^2 \Delta^t + \eta^2 K^2 G^2 + \eta^2 K^2 B^2 \mathbb{E}\|\nabla f(w^t)\|^2 \right) + \frac{\eta^2}{CN} \mathbb{E}\| \sum_{i\in\mathcal{C}} \sum_{k=0}^{K-1} \nabla f_i(w_{i,k}^t)\|^2$$

$$+ \frac{3}{N} \left( 4\eta^2 K^2 L^2 \beta^2 \Delta^t + 3K^3 \eta^4 L^2 \left( \sigma_l^2 + 4KG^2 \right) + 12K^4 \eta^4 L^2 B^2 \mathbb{E}\|\nabla f(w^t)\|^2 \right)$$

$$= \frac{3\beta^2}{N} \left( 1 + 4\eta^2 K^2 L^2 \right) \Delta^t + \frac{\eta^2 K}{N} \left( 1 + 9K^2 \eta^2 L^2 \right) \sigma_l^2 + \frac{3\eta^2 K^2}{N} \left( 1 + 12\eta^2 K^2 L^2 \right) G^2$$

$$+ \frac{3\eta^2 K^2 B^2}{N} \left( 1 + 12\eta^2 K^2 L^2 \right) \mathbb{E}\|\nabla f(w^t)\|^2 + \frac{\eta^2}{CN} \mathbb{E}\| \sum_{i\in\mathcal{C}} \sum_{k=0}^{K-1} \nabla f_i(w_{i,k}^t)\|^2.$$

To minimize the coefficients of each term, we can select a constant order for the term $\eta^2 K^2 L^2$. For convenience, we directly select the $\eta^2 K^2 L^2 \leq 1$ which requires the learning rate $\eta \leq \frac{1}{KL}$. This completes the proof.

**Lemma 4** (Bounded divergence term) *The divergence term $\Delta^t$ could be upper bounded by the local update rules. According to the relaxed initialization in our method, under assumptions stated above, let the learning rate satisfy $\eta \leq \frac{1}{KL}$ and the relaxed coefficient satisfy $\beta \leq \frac{\sqrt{2}}{12}$, the divergence term $\Delta^t$ could be bounded as the recursion of:*

$$\Delta^t \leq \frac{\Delta^t - \Delta^{t+1}}{1 - 72\beta^2} + \frac{51\eta^2 K}{1 - 72\beta^2} \sigma_l^2 + \frac{195\eta^2 K^2}{1 - 72\beta^2} G^2 + \frac{195\eta^2 K^2 B^2}{1 - 72\beta^2} \mathbb{E}\|\nabla f(w^t)\|^2$$

$$+ \frac{3\eta^2}{CN(1 - 72\beta^2)} \mathbb{E}\| \sum_{i\in\mathcal{C}} \sum_{k=0}^{K-1} \nabla f_i(w_{i,k}^t)\|^2. \tag{26}$$

**Proof** *The divergence term measures the inconsistency level in the FL framework. According to the local updates, we have the following recursive formula:*

$$\underbrace{w^{t+1} - w_{i,K}^t}_{local\ bias\ in\ round\ t+1} = \beta \underbrace{(w_{i,K}^{t-1} - w^t)}_{local\ bias\ in\ round\ t} + (w^{t+1} - w^t) + \sum_{k=0}^{K-1} \eta g_{i,k}^t.$$

*By taking the squared norm and expectation on both sides, we have:*

$$\mathbb{E}\|w^{t+1} - w_{i,K}^t\|^2 = \mathbb{E}\|\beta(w_{i,K}^{t-1} - w^t) + w^{t+1} - w^t + \sum_{k=0}^{K-1} \eta g_{i,k}^t\|^2$$

$$\leq 3\beta^2 \mathbb{E}\|w_{i,K}^{t-1} - w^t\|^2 + 3\underbrace{\mathbb{E}\|w^{t+1} - w^t\|^2}_{V_2^t} + 3\mathbb{E}\|\sum_{k=0}^{K-1} \eta g_{i,k}^t\|^2.$$

*The second term in the above inequality is $V_2$ we have bounded in lemma 3. Then we bound the stochastic gradients term. We have:*

$$\mathbb{E}\|\sum_{k=0}^{K-1} \eta g_{i,k}^t\|^2 = \eta^2 \mathbb{E}\|\sum_{k=0}^{K-1} g_{i,k}^t\|^2$$

$$= \eta^2 \mathbb{E}\|\sum_{k=0}^{K-1} \left(g_{i,k}^t - \nabla f_i(w_{i,k}^t)\right)\|^2 + \eta^2 \mathbb{E}\|\sum_{k=0}^{K-1} \nabla f_i(w_{i,k}^t)\|^2$$

$$\leq \eta^2 K \sigma_l^2 + \eta^2 K \sum_{k=0}^{K-1} \mathbb{E}\|\nabla f_i(w_{i,k}^t) - \nabla f_i(w^t) + \nabla f_i(w^t)\|^2$$

$$\leq \eta^2 K \sigma_l^2 + 2\eta^2 K \sum_{k=0}^{K-1} \mathbb{E}\|\nabla f_i(w_{i,k}^t) - \nabla f_i(w^t)\|^2 + 2\eta^2 K \sum_{k=0}^{K-1} \mathbb{E}\|\nabla f_i(w^t)\|^2$$

$$\leq \eta^2 K \sigma_l^2 + 2\eta^2 K L^2 \sum_{k=0}^{K-1} \mathbb{E}\|w_{i,k}^t - w^t\|^2 + 2\eta^2 K^2 \mathbb{E}\|\nabla f_i(w^t)\|^2.$$

*Taking the average on client $i$, we have:*

$$\frac{1}{C}\sum_{i\in\mathcal{C}}\mathbb{E}\|\sum_{k=0}^{K-1} \eta g_{i,k}^t\|^2 \leq \eta^2 K \sigma_l^2 + \frac{2\eta^2 K L^2}{C}\sum_{i\in\mathcal{C}}\sum_{k=0}^{K-1}\mathbb{E}\|w_{i,k}^t - w^t\|^2 + \frac{2\eta^2 K^2}{C}\sum_{i\in\mathcal{C}}\mathbb{E}\|\nabla f_i(w^t)\|^2$$

$$\leq \eta^2 K \sigma_l^2 + 2\eta^2 K L^2 V_1^t + 2\eta^2 K^2 G^2 + 2\eta^2 K^2 B^2 \mathbb{E}\|\nabla f(w^t)\|^2.$$

*Recalling the condition of $\eta \leq \frac{1}{KL}$ and combining this and the squared norm inequality, we have:*

$$\Delta^{t+1} = \frac{1}{C}\sum_{i\in\mathcal{C}}\mathbb{E}\|w^{t+1} - w_{i,K}^t\|^2$$

$$\leq 3\beta^2 \Delta^t + 3V_2^t + \frac{3}{C}\sum_{i\in\mathcal{C}}\mathbb{E}\|\sum_{k=0}^{K-1} \eta g_{i,k}^t\|^2$$

$$\leq 3\beta^2 \left(1 + \frac{15}{N} + 8\eta^2 K^2 L^2\right)\Delta^t + 6\eta^2 K^2 B^2 \left(1 + \frac{39}{2N} + 12\eta^2 K^2 L^2\right)\mathbb{E}\|\nabla f(w^t)\|^2$$

$$+ 3\eta^2 K \left(1 + \frac{10}{N} + 6\eta^2 K^2 L^2\right)\sigma_l^2 + 6\eta^2 K^2 \left(1 + \frac{39}{2N} + 12\eta^2 K^2 L^2\right)G^2$$

$$+ \frac{3\eta^2}{CN}\mathbb{E}\|\sum_{i\in\mathcal{C}}\sum_{k=0}^{K-1} \nabla f_i(w_{i,k}^t)\|^2$$

$$\leq 72\beta^2 \Delta^t + 51\eta^2 K \sigma_l^2 + 195\eta^2 K^2 G^2 + 195\eta^2 K^2 B^2 \mathbb{E}\|\nabla f(w^t)\|^2$$

$$+ \frac{3\eta^2}{CN}\mathbb{E}\|\sum_{i\in\mathcal{C}}\sum_{k=0}^{K-1} \nabla f_i(w_{i,k}^t)\|^2.$$

*Let $72\beta^2 < 1$ where $\beta \leq \frac{\sqrt{2}}{12}$, thus we add $(1 - 72\beta^2)\Delta^t$ on both sides and get the recursive formulation:*

$$(1 - 72\beta^2)\Delta^t \leq (\Delta^t - \Delta^{t+1}) + 51\eta^2 K \sigma_l^2 + 195\eta^2 K^2 G^2 + 195\eta^2 K^2 B^2 \mathbb{E}\|\nabla f(w^t)\|^2$$

$$+ \frac{3\eta^2}{CN}\mathbb{E}\|\sum_{i\in\mathcal{C}}\sum_{k=0}^{K-1} \nabla f_i(w_{i,k}^t)\|^2.$$

*Then we multiply the $\frac{1}{1-72\beta^2}$ on both sides, which completes the proof.*

### A.2.2 Expanding the Smoothness Inequality for the Non-convex Objective

For the non-convex and $L$-smooth function $f$, we firstly expand the smoothness inequality at round $t$ as:

$$\mathbb{E}[f(w^{t+1}) - f(w^t)]$$

$$\leq \mathbb{E}\langle \nabla f(w^t), w^{t+1} - w^t\rangle + \frac{L}{2}\underbrace{\mathbb{E}\|w^{t+1} - w^t\|^2}_{V_2^t}$$

$$= \mathbb{E}\langle \nabla f(w^t), \frac{1}{N}\sum_{i\in\mathcal{N}} w_{i,K}^t - w^t\rangle + \frac{LV_2^t}{2}$$

$$= \mathbb{E}\langle \nabla f(w^t), \frac{1}{C}\sum_{i\in\mathcal{C}} [(w_{i,K}^t - w_{i,0}^t) + \beta(w^t - w_{i,K}^{t-1})]\rangle + \frac{LV_2^t}{2}$$

$$= -\eta\mathbb{E}\langle \nabla f(w^t), \frac{1}{C}\sum_{i\in\mathcal{C}}\sum_{k=0}^{K-1}\nabla f_i(w_{i,k}^t) - \frac{1}{C}\sum_{i\in\mathcal{C}}\sum_{k=0}^{K-1}\nabla f_i(w^t) + K\nabla f(w^t)\rangle + \frac{LV_2^t}{2}$$

$$= -\eta K\mathbb{E}\|f(w^t)\|^2 + \mathbb{E}\langle\sqrt{\eta K}\nabla f(w^t), \sqrt{\frac{\eta}{K}}\frac{1}{C}\sum_{i\in\mathcal{C}}\sum_{k=0}^{K-1}(\nabla f_i(w^t) - \nabla f_i(w_{i,k}^t))\rangle + \frac{LV_2^t}{2}$$

$$\leq -\eta K\mathbb{E}\|f(w^t)\|^2 + \frac{\eta K}{2}\mathbb{E}\|f(w^t)\|^2 + \frac{\eta}{2C}\sum_{i\in\mathcal{C}}\sum_{k=0}^{K-1}\mathbb{E}\|\nabla f_i(w^t) - \nabla f_i(w_{i,k}^t)\|^2$$

$$- \frac{\eta}{2C^2 K}\mathbb{E}\|\sum_{i\in\mathcal{C}}\sum_{k=0}^{K-1}\nabla f_i(w_{i,k}^t)\|^2 + \frac{LV_2^t}{2}$$

$$\leq -\frac{\eta K}{2}\mathbb{E}\|f(w^t)\|^2 + \frac{\eta L^2}{2}\frac{1}{C}\underbrace{\sum_{i\in\mathcal{C}}\sum_{k=0}^{K-1}\mathbb{E}\|w^t - w_{i,k}^t\|^2}_{V_1^t} - \frac{\eta}{2C^2 K}\mathbb{E}\|\sum_{i\in\mathcal{C}}\sum_{k=0}^{K-1}\nabla f_i(w_{i,k}^t)\|^2 + \frac{LV_2^t}{2}$$

$$\leq -\frac{\eta K}{2}\mathbb{E}\|f(w^t)\|^2 + \frac{\eta L^2 V_1^t}{2} - \frac{\eta}{2C^2 K}\mathbb{E}\|\sum_{i\in\mathcal{C}}\sum_{k=0}^{K-1}\nabla f_i(w_{i,k}^t)\|^2 + \frac{LV_2^t}{2}.$$

According to Lemma 2 and lemma 3 to bound the $V_1^t$ and $V_2^t$, we can get the following recursive formula:

$$\mathbb{E}[f(w^{t+1}) - f(w^t)]$$

$$\leq -\frac{\eta K}{2}\mathbb{E}\|f(w^t)\|^2 + \left(\frac{\eta^2 L}{2CN} - \frac{\eta}{2C^2 K}\right)\mathbb{E}\|\sum_{i\in\mathcal{C}}\sum_{k=0}^{K-1}\nabla f_i(w_{i,k}^t)\|^2$$

$$+ \frac{\eta L^2}{2}\left[4K\beta^2\Delta^t + 3K^2\eta^2\left(\sigma_l^2 + 4KG^2\right) + 12K^3\eta^2 B^2\mathbb{E}\|\nabla f(w^t)\|^2\right]$$

$$+ \frac{3\beta^2 L}{2N}\left(1 + 4\eta^2 K^2 L^2\right)\Delta^t + \frac{\eta^2 KL}{2N}\left(1 + 9K^2\eta^2 L^2\right)\sigma_l^2 + \frac{3\eta^2 K^2 L}{2N}\left(1 + 12\eta^2 K^2 L^2\right)G^2$$

$$+ \frac{3\eta^2 K^2 B^2 L}{2N}\left(1 + 12\eta^2 K^2 L^2\right)\mathbb{E}\|\nabla f(w^t)\|^2$$

$$\leq \left(\frac{\eta^2 L}{2CN} - \frac{\eta}{2C^2 K}\right)\mathbb{E}\|\sum_{i\in\mathcal{C}}\sum_{k=0}^{K-1}\nabla f_i(w_{i,k}^t)\|^2 + \frac{3\beta^2 L}{2N}\left[\frac{4N}{3}\eta KL + \left(1 + 4\eta^2 K^2 L^2\right)\right]\Delta^t$$

$$+ \frac{\eta^2 KL}{2N}\left[3N\eta KL + \left(1 + 9\eta^2 K^2 L^2\right)\right]\sigma_l^2 + \frac{3\eta^2 K^2 L}{2N}\left[4N\eta KL + \left(1 + 12\eta^2 K^2 L^2\right)\right]G^2$$

$$- \frac{\eta K}{2}\left[1 - \frac{3\eta KLB^2}{N}\left(1 + 12\eta^2 K^2 L^2\right) - 12\eta^2 K^2 L^2 B^2\right]\mathbb{E}\|f(w^t)\|^2.$$

Here we make a comprehensive discussion on the selection of $\eta$ to simplify the above formula. In fact, in lemma 2, there is a constraint on the learning rate as $\eta \leq \frac{\sqrt{2}}{4(K-1)L}$ for $K \geq 2$. In lemma 3 and lemma 4, there is a constraint on the learning rate as $\eta \leq \frac{1}{KL}$. To further minimize the coefficient, we select the $N\eta KL$ to be constant order. For convenience, we directly select the $\eta \leq \frac{1}{NKL}$. Thus, we have:

$$\mathbb{E}[f(w^{t+1}) - f(w^t)]$$

$$\leq \frac{3\beta^2 L}{2N}\left(\frac{4}{3}N\eta KL + 5\right)\Delta^t + \frac{\eta^2 KL}{2N}\left(3N\eta KL + 10\right)\sigma_l^2 + \frac{3\eta^2 K^2 L}{2N}\left(4N\eta KL + 13\right)G^2$$

$$-\frac{\eta K}{2}\left(1-\frac{39\eta KLB^2}{N}-12\eta^2K^2L^2B^2\right)\mathbb{E}\|f(w^t)\|^2$$

$$+\left(\frac{\eta^2 L}{2CN}-\frac{\eta}{2C^2K}\right)\mathbb{E}\|\sum_{i\in\mathcal{C}}\sum_{k=0}^{K-1}\nabla f_i(w_{i,k}^t)\|^2$$

$$<\frac{10\beta^2 L(\Delta^t-\Delta^{t+1})}{(1-72\beta^2)N}+\frac{3\eta^2K^2L}{2N}\left(\frac{1300\beta^2}{1-72\beta^2}+17\right)G^2+\frac{\eta^2KL}{2N}\left(\frac{1020\beta^2}{1-72\beta^2}+13\right)\sigma_l^2$$

$$+\left[\frac{30\beta^2\eta^2 L}{CN^2(1-72\beta^2)}+\frac{\eta^2 L}{2CN}-\frac{\eta}{2C^2K}\right]\mathbb{E}\|\sum_{i\in\mathcal{C}}\sum_{k=0}^{K-1}\nabla f_i(w_{i,k}^t)\|^2$$

$$-\frac{\eta K}{2}\left[1-\frac{39\eta KLB^2}{N}-\frac{3900\beta^2\eta KLB^2}{(1-72\beta^2)N}-12\eta^2K^2L^2B^2\right]\mathbb{E}\|f(w^t)\|^2.$$

Firstly, to remove the gradient term, we follow the [19, 48] and let $\frac{30\beta^2\eta^2 L}{CN^2(1-72\beta^2)}+\frac{\eta^2 L}{2CN}-\frac{\eta}{2C^2K}\leq 0$, then learning rate $\eta\leq\frac{N}{2CKL}$. Then, according to the [48], there is a positive constant $\lambda\in(0,1)$ to satisfy $1-\frac{39\eta KLB^2}{N}-\frac{3900\beta^2\eta KLB^2}{(1-72\beta^2)N}-12\eta^2K^2L^2B^2>\lambda>0$. We denote $\kappa_1=\frac{1300\beta^2}{1-72\beta^2}+17$ and $\kappa_2=\frac{1020\beta^2}{1-72\beta^2}+13$ as two constants in the formula. Therefore, we have:

$$\frac{\lambda\eta K}{2}\mathbb{E}\|f(w^t)\|^2$$

$$\leq\mathbb{E}[f(w^t)-f(w^{t+1})]+\frac{10\beta^2 L}{(1-72\beta^2)N}(\Delta^t-\Delta^{t+1})+\frac{3\kappa_1\eta^2K^2L}{2N}G^2+\frac{\kappa_2\eta^2KL}{2N}\sigma_l^2.$$

### A.2.3 Proof of Theorem 1

**Theorem 7** *Under Assumption 6~8, let participation ratio is $N/C$ where $1<N<C$, let the learning rate satisfy $\eta\leq\min\left\{\frac{N}{2CKL},\frac{1}{NKL}\right\}$ where $K\geq 2$, let the relaxation coefficient $\beta\leq\frac{\sqrt{2}}{12}$, and after training $T$ rounds, the global model $w^t$ generated by FedInit satisfies:*

$$\frac{1}{T}\sum_{t=0}^{T-1}\mathbb{E}\|f(w^t)\|^2\leq\frac{2\left(f(w^0)-f(w^\star)\right)}{\lambda\eta K}+\frac{\kappa_2\eta L}{\lambda N}\sigma_l^2+\frac{3\kappa_1\eta KL}{\lambda N}G^2. \tag{27}$$

*where $\lambda\in(0,1)$, $\kappa_1=\frac{1300\beta^2}{1-72\beta^2}+17$, and $\kappa_2=\frac{1020\beta^2}{1-72\beta^2}+13$ are three constants.*

*Further, by selecting the proper learning rate $\eta=\mathcal{O}(\sqrt{\frac{N}{KT}})$ and let $D=f(w^0)-f(w^\star)$ as the initialization bias, the global model $w^t$ satisfies:*

$$\frac{1}{T}\sum_{t=0}^{T-1}\mathbb{E}\|f(w^t)\|^2=\mathcal{O}\left(\frac{D+L\left(\sigma_l^2+3KG^2\right)}{\sqrt{NKT}}\right). \tag{28}$$

**Proof** *According to the expansion of the smoothness inequality, we have:*

$$\frac{\lambda\eta K}{2}\mathbb{E}\|f(w^t)\|^2$$

$$\leq\mathbb{E}[f(w^t)-f(w^{t+1})]+\frac{10\beta^2 L}{(1-72\beta^2)N}(\Delta^t-\Delta^{t+1})+\frac{3\kappa_1\eta^2K^2L}{2N}G^2+\frac{\kappa_2\eta^2KL}{2N}\sigma_l^2.$$

*Taking the accumulation from 0 to $T-1$, we have:*

$$\frac{1}{T}\sum_{t=0}^{T-1}\mathbb{E}\|f(w^t)\|^2$$

$$\leq\frac{2\mathbb{E}[f(w^0)-f(w^T)]}{\lambda\eta KT}+\frac{20\beta^2 L}{(1-72\beta^2)\lambda\eta KNT}(\Delta^0-\Delta^T)+\frac{\kappa_2\eta L}{\lambda N}\sigma_l^2+\frac{3\kappa_1\eta KL}{\lambda N}G^2$$

$$\leq\frac{2\left(f(w^0)-f(w^\star)\right)}{\lambda\eta KT}+\frac{\kappa_2\eta L}{\lambda N}\sigma_l^2+\frac{3\kappa_1\eta KL}{\lambda N}G^2.$$

*We select the learning rate $\eta=\mathcal{O}(\sqrt{\frac{N}{KT}})$ and let $D=f(w^0)-f(w^\star)$ as the initialization bias, then we have:*

$$\frac{1}{T}\sum_{t=0}^{T-1}\mathbb{E}\|f(w^t)\|^2=\mathcal{O}\left(\frac{D+L\left(\sigma_l^2+3KG^2\right)}{\sqrt{NKT}}\right).$$

*This completes the proof.*

### A.2.4 Proof of Theorem 2

**Theorem 8** *Under Assumption 6∼8 and 10, let participation ratio is $N/C$ where $1 < N < C$, let the learning rate satisfy $\eta \le \min\left\{\frac{N}{2CKL}, \frac{1}{NKL}, \frac{1}{\lambda\mu K}\right\}$ where $K \ge 2$, let the relaxation coefficient $\beta \le \frac{\sqrt{2}}{12}$, and after training $T$ rounds, the global model $w^t$ generated by FedInit satisfies:*

$$\mathbb{E}[f(w^T) - f(w^\star)] \le e^{-\lambda\mu\eta KT}\mathbb{E}[f(w^0) - f(w^\star)] + \frac{3\kappa_1\eta KL}{2N\lambda\mu}G^2 + \frac{\kappa_2\eta L}{2N\lambda\mu}\sigma_l^2. \tag{29}$$

*Further, by selecting the proper learning rate $\eta = \mathcal{O}\left(\frac{\log(\lambda\mu NKT)}{\lambda\mu KT}\right)$ and let $D = f(w^0) - f(w^\star)$ as the initialization bias, the global model $w^t$ satisfies:*

$$\mathbb{E}[f(w^T) - f(w^\star)] = \mathcal{O}\left(\frac{D + L(\sigma_l^2 + KG^2)}{NKT}\right). \tag{30}$$

**Proof** *According to the expansion of the smoothness inequality, we have:*

$$\frac{\lambda\eta K}{2}\mathbb{E}\|f(w^t)\|^2$$

$$\le \mathbb{E}[f(w^t) - f(w^{t+1})] + \frac{10\beta^2 L}{(1 - 72\beta^2)N}(\Delta^t - \Delta^{t+1}) + \frac{3\kappa_1\eta^2 K^2 L}{2N}G^2 + \frac{\kappa_2\eta^2 KL}{2N}\sigma_l^2.$$

*According to Assumption 10, we have $2\mu(f(w) - f(w^\star)) \le \|\nabla f(w)\|^2$, we have:*

$$\lambda\mu\eta K\mathbb{E}[f(w^t) - f(w^\star)] \le \frac{\lambda\eta K}{2}\mathbb{E}\|f(w^t)\|^2$$

$$\le \mathbb{E}[f(w^t) - f(w^{t+1})] + \frac{10\beta^2 L}{(1 - 72\beta^2)N}(\Delta^t - \Delta^{t+1}) + \frac{3\kappa_1\eta^2 K^2 L}{2N}G^2 + \frac{\kappa_2\eta^2 KL}{2N}\sigma_l^2.$$

*Combining the terms aligned with $w^t$ and $w^{t+1}$, we have:*

$$\mathbb{E}[f(w^{t+1}) - f(w^\star)]$$

$$\le (1 - \lambda\mu\eta K)\mathbb{E}[f(w^t) - f(w^\star)] + \frac{10\beta^2 L}{(1 - 72\beta^2)N}(\Delta^t - \Delta^{t+1}) + \frac{3\kappa_1\eta^2 K^2 L}{2N}G^2 + \frac{\kappa_2\eta^2 KL}{2N}\sigma_l^2.$$

*Taking the recursion from $t = 0$ to $T - 1$ and let learning rate $\eta \le \frac{1}{\lambda\mu K}$, we have:*

$$\mathbb{E}[f(w^T) - f(w^\star)]$$

$$\le (1 - \lambda\mu\eta K)^T\mathbb{E}[f(w^0) - f(w^\star)] + \sum_{t=0}^{T-1}(1 - \lambda\mu\eta K)^{T-1-t}\frac{10\beta^2 L}{(1 - 72\beta^2)N}(\Delta^t - \Delta^{t+1})$$

$$+ \left(\frac{3\kappa_1\eta^2 K^2 L}{2N}G^2 + \frac{\kappa_2\eta^2 KL}{2N}\sigma_l^2\right)\sum_{t=0}^{T-1}(1 - \lambda\mu\eta K)^{T-1-t}$$

$$\le (1 - \lambda\mu\eta K)^T\mathbb{E}[f(w^0) - f(w^\star)] + \frac{10\beta^2 L}{(1 - 72\beta^2)N}(\Delta^0 - \Delta^T)$$

$$+ \left(\frac{3\kappa_1\eta^2 K^2 L}{2N}G^2 + \frac{\kappa_2\eta^2 KL}{2N}\sigma_l^2\right)\frac{1 - (1 - \lambda\mu\eta K)^T}{\lambda\mu\eta K}$$

$$\le (1 - \lambda\mu\eta K)^T\mathbb{E}[f(w^0) - f(w^\star)] + \frac{3\kappa_1\eta KL}{2N\lambda\mu}G^2 + \frac{\kappa_2\eta L}{2N\lambda\mu}\sigma_l^2$$

$$\le e^{-\lambda\mu\eta KT}\mathbb{E}[f(w^0) - f(w^\star)] + \frac{3\kappa_1\eta KL}{2N\lambda\mu}G^2 + \frac{\kappa_2\eta L}{2N\lambda\mu}\sigma_l^2.$$

*We select the learning rate $\eta = \mathcal{O}\left(\frac{\log(\lambda\mu NKT)}{\lambda\mu KT}\right)$ and let $D = f(w^0) - f(w^\star)$ as the initialization bias, then we have:*

$$\mathbb{E}[f(w^T) - f(w^\star)] = \mathcal{O}\left(\frac{D + L(\sigma_l^2 + KG^2)}{NKT}\right).$$

*This completes the proof.*

### A.2.5 Proof of Theorem 4

**Theorem 9** *Under Assumption 6∼8, we can bound the divergence term as follows. Let the learning rate satisfy $\eta \le \min\left\{\frac{N}{2CKL}, \frac{1}{NKL}, \frac{\sqrt{N}}{\sqrt{C}KL}\right\}$ where $K \ge 2$, and after training $T$ rounds, let $0 < \beta < \frac{\sqrt{6}}{24}$, the divergence*

*term $\Delta^t$ generated by FedInit satisfies:*

$$\frac{1}{T}\sum_{t=0}^{T-1}\Delta^t = \mathcal{O}\left(\frac{N(\sigma_l^2 + KG^2)}{T} + \frac{\sqrt{NK}B^2\left[D + L(\sigma_l^2 + KG^2)\right]}{T^{\frac{3}{2}}}\right). \tag{31}$$

**Proof** *According to Lemma 4, we have:*

$$\Delta^{t+1} \leq 72\beta^2\Delta^t + 51\eta^2K\sigma_l^2 + 195\eta^2K^2G^2 + 195\eta^2K^2B^2\mathbb{E}\|\nabla f(w^t)\|^2$$
$$+ \frac{3\eta^2}{CN}\mathbb{E}\|\sum_{i\in\mathcal{C}}\sum_{k=0}^{K-1}\nabla f_i(w_{i,k}^t)\|^2.$$

*Here we further bound the gradient term, we have:*

$$\mathbb{E}\|\sum_{i\in\mathcal{C}}\sum_{k=0}^{K-1}\nabla f_i(w_{i,k}^t)\|^2 = \mathbb{E}\|\sum_{i\in\mathcal{C}}\sum_{k=0}^{K-1}\left(\nabla f_i(w_{i,k}^t) - \nabla f_i(w^t) + \nabla f_i(w^t)\right)\|^2$$

$$= \mathbb{E}\|\sum_{i\in\mathcal{C}}\sum_{k=0}^{K-1}\left(\nabla f_i(w_{i,k}^t) - \nabla f_i(w^t) + \nabla f(w^t)\right)\|^2$$

$$\leq CK\sum_{i\in\mathcal{C}}\sum_{k=0}^{K-1}\mathbb{E}\|\nabla f_i(w_{i,k}^t) - \nabla f_i(w^t) + \nabla f(w^t)\|^2$$

$$\leq 2CK\sum_{i\in\mathcal{C}}\sum_{k=0}^{K-1}\mathbb{E}\|\nabla f_i(w_{i,k}^t) - \nabla f_i(w^t)\|^2 + 2C^2K^2\mathbb{E}\|\nabla f(w^t)\|^2$$

$$\leq 2C^2KL^2V_1^t + 2C^2K^2\mathbb{E}\|\nabla f(w^t)\|^2.$$

*Combining this into the recursive formulation, and let the learning rate satisfy $\eta \leq \frac{\sqrt{N}}{\sqrt{C}KL}$, we have:*

$$\Delta^{t+1} \leq \beta^2\left(72 + \frac{24C\eta^2K^2L^2}{N}\right)\Delta^t + \eta^2K^2B^2\left(195 + \frac{72C\eta^2K^2L^2}{N}\right)\mathbb{E}\|\nabla f(w^t)\|^2$$
$$+ \eta^2K\left(51 + \frac{18C\eta^2K^2L^2}{N}\right)\sigma_l^2 + \eta^2K^2\left(195 + \frac{72C\eta^2K^2L^2}{N}\right)G^2$$
$$\leq 96\beta^2\Delta^t + 267\eta^2K^2B^2\mathbb{E}\|\nabla f(w^t)\|^2 + 69\eta^2K\sigma_l^2 + 267\eta^2K^2G^2.$$

*Let $96\beta^2 < 1$ as the decayed coefficient where $\beta < \frac{\sqrt{6}}{24}$, similar as Lemma 4, we have:*

$$\Delta^t \leq \frac{\Delta^t - \Delta^{t+1}}{1 - 96\beta^2} + \frac{267\eta^2K^2B^2}{1 - 96\beta^2}\mathbb{E}\|\nabla f(w^t)\|^2 + \frac{69\eta^2K}{1 - 96\beta^2}\sigma_l^2 + \frac{267\eta^2K^2}{1 - 96\beta^2}G^2.$$

*by taking the accumulation from $t = 0$ to $T - 1$,*

$$\frac{1}{T}\sum_{t=0}^{T-1}\Delta^t \leq \frac{\Delta^0 - \Delta^T}{1 - 96\beta^2} + \frac{69\eta^2K}{1 - 96\beta^2}\sigma_l^2 + \frac{267\eta^2K^2}{1 - 96\beta^2}G^2 + \frac{267\eta^2K^2B^2}{1 - 96\beta^2}\frac{1}{T}\sum_{t=0}^{T-1}\|\nabla f(w^t)\|^2$$

$$\leq \frac{267\eta^2K^2B^2}{1 - 96\beta^2}\left(\frac{2\left(f(w^0) - f(w^\star)\right)}{\lambda\eta KT} + \frac{\kappa_2\eta L}{\lambda N}\sigma_l^2 + \frac{3\kappa_1\eta KL}{\lambda N}G^2\right)$$

$$+ \frac{69\eta^2K}{1 - 96\beta^2}\sigma_l^2 + \frac{267\eta^2K^2}{1 - 96\beta^2}G^2$$

$$\leq \frac{534\eta KB^2\left(f(w^0) - f(w^\star)\right)}{(1 - 96\beta^2)\lambda T} + \frac{267\eta^3K^2B^2\kappa_2L}{(1 - 96\beta^2)\lambda N}\sigma_l^2 + \frac{801\eta^3K^3B^2\kappa_1L}{(1 - 96\beta^2)\lambda N}G^2$$

$$+ \frac{69\eta^2K}{1 - 96\beta^2}\sigma_l^2 + \frac{267\eta^2K^2}{1 - 96\beta^2}G^2.$$

*The same, the learning rate is selected as $\eta = \mathcal{O}(\sqrt{\frac{N}{KT}})$ and let $D = f(w^0) - f(w^\star)$ as the initialization bias and let $96\beta^2 < 1$, thus we have:*

$$\frac{1}{T}\sum_{t=0}^{T-1}\Delta^t = \mathcal{O}\left(\frac{N(\sigma_l^2 + KG^2)}{T} + \frac{\sqrt{NK}B^2\left[D + L(\sigma_l^2 + KG^2)\right]}{T^{\frac{3}{2}}}\right).$$

*This completes this proof.*

### A.2.6 Proof of Theorem 5

**Theorem 10** *Under Assumption 6∼8 and 10, we can bound the divergence term as follows. Let the learning rate satisfy $\eta \leq \min\left\{\frac{N}{2CKL}, \frac{1}{NKL}, \frac{1}{\lambda\mu K}\right\}$ where $K \geq 2$, and after training $T$ rounds, let $0 < \beta < \frac{\sqrt{6}}{24}$, the divergence term $\Delta^T$ generated by FedInit satisfies:*

$$\Delta^T = \mathcal{O}\left(\frac{D + G^2}{T^2} + \frac{N\sigma_l^2 + KG^2}{NKT^2}\right) + \mathcal{O}\left(\frac{1}{NKT^3}\right). \tag{32}$$

**Proof** *According to the Theorem 8, we have:*

$$\Delta^{t+1} \leq 96\beta^2\Delta^t + 267\eta^2 K^2 B^2 \mathbb{E}\|\nabla f(w^t)\|^2 + 69\eta^2 K\sigma_l^2 + 267\eta^2 K^2 G^2.$$

*Taking the recursive formulation from $t = 0$ to $T - 1$, we have:*

$$\Delta^T \leq (96\beta^2)^T\Delta^0 + \sum_{t=0}^{T-1}(96\beta^2)^t\left(267\eta^2 K^2 B^2 \mathbb{E}\|\nabla f(w^t)\|^2 + 69\eta^2 K\sigma_l^2 + 267\eta^2 K^2 G^2\right)$$

$$\leq \frac{69\eta^2 K\sigma_l^2}{1 - 96\beta^2} + \frac{267\eta^2 K^2 G^2}{1 - 96\beta^2} + 267\eta^2 K^2 B^2 \sum_{t=0}^{T-1}(96\beta^2)^{T-1-t}\mathbb{E}\|\nabla f(w^t)\|^2$$

$$\leq \frac{69\eta^2 K\sigma_l^2}{1 - 96\beta^2} + \frac{267\eta^2 K^2 G^2}{1 - 96\beta^2} + \frac{267\eta^2 K^2 B^2}{1 - 96\beta^2}\left(\frac{\kappa_2\eta L}{\lambda N}\sigma_l^2 + \frac{3\kappa_1\eta KL}{\lambda N}G^2\right)$$

$$+ \frac{534\eta KB^2}{\lambda}\sum_{t=0}^{T-1}(96\beta^2)^{T-1-t}\mathbb{E}\left[f(w^t) - f(w^{t+1})\right] + \frac{10\beta^2 L}{(1 - 72\beta^2)N}(\Delta^0 - \Delta^T)$$

$$\leq \frac{69\eta^2 K\sigma_l^2}{1 - 96\beta^2} + \frac{267\eta^2 K^2 G^2}{1 - 96\beta^2} + \frac{267B^2 L}{(1 - 96\beta^2)\lambda}\left(\frac{\kappa_2\eta^3 K^2}{N}\sigma_l^2 + \frac{3\kappa_1\eta^3 K^3}{N}G^2\right)$$

$$+ \frac{534\eta KB^2}{\lambda}\sum_{t=0}^{T-1}(96\beta^2)^{T-1-t}\mathbb{E}\left[f(w^t) - f(w^\star)\right].$$

*According to the Theorem 11, we have:*

$$\mathbb{E}[f(w^t) - f(w^\star)] \leq e^{-\lambda\mu\eta Kt}\mathbb{E}[f(w^0) - f(w^\star)] + \frac{3\kappa_1\eta KL}{2N\lambda\mu}G^2 + \frac{\kappa_2\eta L}{2N\lambda\mu}\sigma_l^2.$$

*Let $96\beta^2 \leq e^{-\lambda\mu\eta K}$, thus we have:*

$$\frac{534\eta KB^2}{\lambda}\sum_{t=0}^{T-1}(96\beta^2)^{T-1-t}\mathbb{E}\left[f(w^t) - f(w^\star)\right]$$

$$\leq \frac{267B^2 L}{(1 - 96\beta^2)\lambda^2\mu}\left(\frac{\kappa_2\eta^2 K}{N}\sigma_l^2 + \frac{3\kappa_1\eta^2 K^2}{N}G^2\right) + \frac{534\eta KB^2}{\lambda}\mathbb{E}[f(w^0) - f(w^\star)]\sum_{t=0}^{T-1}(96\beta^2)^{T-1-t}e^{-\lambda\mu\eta Kt}$$

$$\leq \frac{267B^2 L}{(1 - 96\beta^2)\lambda^2\mu}\left(\frac{\kappa_2\eta^2 K}{N}\sigma_l^2 + \frac{3\kappa_1\eta^2 K^2}{N}G^2\right)$$

$$+ \frac{534\eta KB^2}{\lambda}\mathbb{E}[f(w^0) - f(w^\star)]e^{-\lambda\mu\eta KT}\sum_{t=0}^{T-1}e^{-2\lambda\mu\eta Kt}.$$

*Thus selecting the same learning rate $\eta = \mathcal{O}\left(\frac{\log(\lambda\mu NKT)}{\lambda\mu KT}\right)$ and let $D = f(w^0) - f(w^\star)$ as the initialization bias, we have:*

$$\Delta^T = \mathcal{O}\left(\frac{D + G^2}{T^2} + \frac{N\sigma_l^2 + KG^2}{NKT^2} + \frac{1}{NKT^3}\right).$$

*This completes the proof.*

### A.3 Proofs for the Generalization Error

In this part, we prove the generalization error for our proposed method. We assume the objective function $f$ is $L$-smooth and $L_G$-Lipschitz as defined in [11, 51]. We follow the uniform stability to upper bound the generalization error in the FL.

We suppose there are $C$ clients participating in the training process as a set $\mathcal{C} = \{i\}_{i=1}^C$. Each client has a local dataset $\mathcal{S}_i = \{z_j\}_{j=1}^S$ with total $S$ data sampled from a specific unknown distribution $\mathcal{D}_i$. Now we define a re-sampled dataset $\widetilde{\mathcal{S}}_i$ which only differs from the dataset $\mathcal{S}_i$ on the $j^\star$-th data. We replace the $\mathcal{S}_{i^\star}$ with $\widetilde{\mathcal{S}}_{i^\star}$ and keep other $C-1$ local dataset, which composes a new set $\widetilde{\mathcal{C}}$. From the perspective of total data, $\mathcal{C}$ only differs from the $\widetilde{\mathcal{C}}$ at $j^\star$-th data on the $i^\star$-th client. Then, based on these two sets, our method could generate two output models, $w^t$ and $\widetilde{w}^t$ respectively, after $t$ training rounds. We first introduce some notations used in the proof of the generalization error.

Table 5: Some abbreviations of the used terms in the proof of bounded training error.

| Notation | Formulation | Description |
|---|---|---|
| $w$ | - | parameters trained with set $\mathcal{C}$ |
| $\widetilde{w}$ | - | parameters trained with set $\widetilde{\mathcal{C}}$ |
| $\Delta^t$ | $\frac{1}{C}\sum_{i\in\mathcal{C}}\mathbb{E}\|w_{i,K}^{t-1} - w^t\|^2$ | inconsistency/divergence term in round $t$ |

Then we introduce some important lemmas in our proofs.

#### A.3.1 Important Lemmas

**Lemma 5** (Lemma 3.11 in [11]) *We follow the definition in [11, 51] to upper bound the uniform stability term after each communication round in FL paradigm. Different from their vanilla calculations, FL considers the finite-sum function on heterogeneous clients. Let non-negative objective $f$ is $L$-smooth and $L_G$-Lipschitz. After training $T$ rounds on $\mathcal{C}$ and $\widetilde{\mathcal{C}}$, our method generates two models $w^{T+1}$ and $\widetilde{w}^{T+1}$ respectively. For each data $z$ and every $t_0 \in \{1, 2, 3, \cdots, S\}$, we have:*

$$\mathbb{E}\|f(w^{T+1};z) - f(\widetilde{w}^{T+1};z)\| \le \frac{Ut_0}{S} + \frac{L_G}{C}\sum_{i\in\mathcal{C}}\mathbb{E}\left[\|w_{i,K}^T - \widetilde{w}_{i,K}^T\| \mid \xi\right]. \tag{33}$$

**Proof** *Let $\xi = 1$ denote the event $\|w^{t_0} - \widetilde{w}^{t_0}\| = 0$ and $U = \sup_{w,z} f(w;z)$, we have:*

$$\mathbb{E}\|f(w^{T+1};z) - f(\widetilde{w}^{T+1};z)\|$$
$$= P(\{\xi\})\,\mathbb{E}\left[\|f(w^{T+1};z) - f(\widetilde{w}^{T+1};z)\| \mid \xi\right] + P(\{\xi^c\})\,\mathbb{E}\left[\|f(w^{T+1};z) - f(\widetilde{w}^{T+1};z)\| \mid \xi^c\right]$$
$$\le \mathbb{E}\left[\|f(w^{T+1};z) - f(\widetilde{w}^{T+1};z)\| \mid \xi\right] + P(\{\xi^c\})\sup_{w,z} f(w;z)$$
$$\le L_G\mathbb{E}\left[\|w^{T+1} - \widetilde{w}^{T+1}\| \mid \xi\right] + UP(\{\xi^c\})$$
$$= L_G\mathbb{E}\left[\|\frac{1}{C}\sum_{i\in\mathcal{C}}(w_{i,K}^T - \widetilde{w}_{i,K}^T)\| \mid \xi\right] + UP(\{\xi^c\})$$
$$\le \frac{L_G}{C}\sum_{i\in\mathcal{C}}\mathbb{E}\left[\|w_{i,K}^T - \widetilde{w}_{i,K}^T\| \mid \xi\right] + UP(\{\xi^c\}).$$

*Before the $j^\star$-th data on $i^\star$-th client is sampled, the iterative states are identical on both $\mathcal{C}$ and $\widetilde{\mathcal{C}}$. Let $\widetilde{j}$ is the index of the first different sampling, if $\widetilde{j} > t_0$, then $\xi = 1$ hold for $t_0$. Therefore, we have:*

$$P(\{\xi^c\}) = P(\{\xi = 0\}) \le P(\widetilde{j} \le t_0) \le \frac{t_0}{S},$$

*where $\widetilde{j}$ is uniformly selected. This completes the proof.*

**Lemma 6** (Lemma 1.1 in [51]) *Different from their calculations, we prove the similar inequalities on $f$ in the stochastic optimization. Let non-negative objective $f$ is $L$-smooth w.r.t $w$. The local updates satisfy $w_{i,k+1}^t = w_{i,k}^t - \eta g_{i,k}^t$ on $\mathcal{C}$ and $\widetilde{w}_{i,k+1}^t = \widetilde{w}_{i,k}^t - \eta\widetilde{g}_{i,k}^t$ on $\widetilde{\mathcal{C}}$. If at $k$-th iteration on each round, we sample the **same** data in $\mathcal{C}$ and $\widetilde{\mathcal{C}}$, then we have:*

$$\mathbb{E}\|w_{i,k+1}^t - \widetilde{w}_{i,k+1}^t\| \le (1 + \eta L)\mathbb{E}\|w_{i,k}^t - \widetilde{w}_{i,k}^t\| + 2\eta\sigma_l. \tag{34}$$

**Proof** *In each round $t$, by the triangle inequality and omitting the same data $z$, we have:*

$$\mathbb{E}\|w_{i,k+1}^t - \widetilde{w}_{i,k+1}^t\|$$
$$= \mathbb{E}\|w_{i,k}^t - \eta g_{i,k}^t - \widetilde{w}_{i,k}^t - \eta \widetilde{g}_{i,k}^t\|$$
$$\leq \mathbb{E}\|w_{i,k}^t - \widetilde{w}_{i,k}^t\| + \eta \mathbb{E}\|g_{i,k}^t - \widetilde{g}_{i,k}^t\|$$
$$= \mathbb{E}\|w_{i,k}^t - \widetilde{w}_{i,k}^t\| + \eta \mathbb{E}\| \left(g_{i,k}^t - \nabla f_i(w_{i,k}^t)\right) - \left(\widetilde{g}_{i,k}^t - \nabla f_i(\widetilde{w}_{i,k}^t)\right) + \left(\nabla f_i(w_{i,k}^t) - \nabla f_i(\widetilde{w}_{i,k}^t)\right)\|$$
$$\leq \mathbb{E}\|w_{i,k}^t - \widetilde{w}_{i,k}^t\| + \eta \mathbb{E}\|g_{i,k}^t - \nabla f_i(w_{i,k}^t)\| + \eta \mathbb{E}\|\widetilde{g}_{i,k}^t - \nabla f_i(\widetilde{w}_{i,k}^t)\| + \eta \mathbb{E}\|\nabla f_i(w_{i,k}^t) - \nabla f_i(\widetilde{w}_{i,k}^t)\|$$
$$\leq (1 + \eta L)\mathbb{E}\|w_{i,k}^t - \widetilde{w}_{i,k}^t\| + 2\eta \sigma_l.$$

*The final inequality adopts assumptions of $\mathbb{E}\|g_{i,k}^t - \nabla f_i(w_{i,k}^t)\| \leq \sqrt{\mathbb{E}\|g_{i,k}^t - \nabla f_i(w_{i,k}^t)\|^2} \leq \sigma_l$. This completes the proof.*

**Lemma 7** (Lemma 1.2 in [51]) *Different from their calculations, we prove the similar inequalities on $f$ in the stochastic optimization. Let non-negative objective $f$ is $L$-smooth and $L_G$-Lipschitz w.r.t $w$. The local updates satisfy $w_{i,k+1}^t = w_{i,k}^t - \eta g_{i,k}^t$ on $\mathcal{C}$ and $\widetilde{w}_{i,k+1}^t = \widetilde{w}_{i,k}^t - \eta \widetilde{g}_{i,k}^t$ on $\widetilde{\mathcal{C}}$. If at $k$-th iteration on each round, we sample the **different** data in $\mathcal{C}$ and $\widetilde{\mathcal{C}}$, then we have:*

$$\mathbb{E}\|w_{i,k+1}^t - \widetilde{w}_{i,k+1}^t\| \leq \mathbb{E}\|w_{i,k}^t - \widetilde{w}_{i,k}^t\| + 2\eta(\sigma_l + L_G). \tag{35}$$

**Proof** *In each round $t$, let by the triangle inequality and denoting the different data as $z$ and $\widetilde{z}$, we have:*

$$\mathbb{E}\|w_{i,k+1}^t - \widetilde{w}_{i,k+1}^t\|$$
$$= \mathbb{E}\|w_{i,k}^t - \eta g_{i,k}^t - \widetilde{w}_{i,k}^t - \eta \widetilde{g}_{i,k}^t\|$$
$$\leq \mathbb{E}\|w_{i,k}^t - \widetilde{w}_{i,k}^t\| + \eta \mathbb{E}\|g_{i,k}^t - \widetilde{g}_{i,k}^t\|$$
$$= \mathbb{E}\|w_{i,k}^t - \widetilde{w}_{i,k}^t\| + \eta \mathbb{E}\|g_{i,k}^t - \nabla f_i(w_{i,k}^t; z) - \widetilde{g}_{i,k}^t - \nabla f_i(\widetilde{w}_{i,k}^t; \widetilde{z}) + \nabla f_i(w_{i,k}^t; z) - \nabla f_i(\widetilde{w}_{i,k}^t; \widetilde{z})\|$$
$$\leq \mathbb{E}\|w_{i,k}^t - \widetilde{w}_{i,k}^t\| + 2\eta \sigma_l + \eta \mathbb{E}\|\nabla f_i(w_{i,k}^t; z) - \nabla f_i(\widetilde{w}_{i,k}^t; \widetilde{z})\|$$
$$\leq \mathbb{E}\|w_{i,k}^t - \widetilde{w}_{i,k}^t\| + 2\eta(\sigma_l + L_G).$$

*The final inequality adopts the $L_G$-Lipschitz. This completes the proof.*

### A.3.2 Bounded Uniform Stability

According to Lemma 5, we firstly bound the recursive stability on $k$ in one round. If the sampled data is the same, we can adopt Lemma 6. Otherwise, we adopt Lemma 7. Thus we can bound the second term in Lemma 5 as:

$$\mathbb{E}\left[\|w_{i,k+1}^t - \widetilde{w}_{i,k+1}^t\| \mid \xi\right]$$
$$= P(z)\, \mathbb{E}\left[\|w_{i,k+1}^t - \widetilde{w}_{i,k+1}^t\| \mid \xi, z\right] + P(\widetilde{z})\, \mathbb{E}\left[\|w_{i,k+1}^t - \widetilde{w}_{i,k+1}^t\| \mid \xi, \widetilde{z}\right]$$
$$\leq \left(1 - \frac{1}{S}\right)(1 + \eta L)\mathbb{E}\left[\|w_{i,k}^t - \widetilde{w}_{i,k}^t\| \mid \xi\right] + 2\eta \sigma_l + \frac{1}{S}\mathbb{E}\left[\|w_{i,k}^t - \widetilde{w}_{i,k}^t\| \mid \xi\right] + \frac{2\eta L_G}{S}$$
$$= \left(1 + \left(1 - \frac{1}{S}\right)\eta L\right)\mathbb{E}\left[\|w_{i,k}^t - \widetilde{w}_{i,k}^t\| \mid \xi\right] + \frac{2\eta L_G}{S} + 2\eta \sigma_l$$
$$\leq e^{\left(1 - \frac{1}{S}\right)\eta L}\mathbb{E}\left[\|w_{i,k}^t - \widetilde{w}_{i,k}^t\| \mid \xi\right] + \frac{2\eta L_G}{S} + 2\eta \sigma_l.$$

At the beginning of each round $t$, FL paradigm will aggregate the last state of each client $w_{i,K}^{t-1}$, according to our method, $w_{i,0}^t = w^t + \beta(w^t - w_{i,K}^{t-1})$, thus the relationship between them is:

$$\frac{1}{C}\sum_{i\in\mathcal{C}}\mathbb{E}\|w_{i,0}^t - w_{i,K}^{t-1}\| = (1 + \beta)\frac{1}{C}\sum_{i\in\mathcal{C}}\mathbb{E}\|w^t - w_{i,K}^{t-1}\| \leq (1 + \beta)\frac{1}{C}\sum_{i\in\mathcal{C}}\sqrt{\mathbb{E}\|w^t - w_{i,K}^{t-1}\|^2}$$
$$\leq (1 + \beta)\sqrt{\frac{1}{C}\sum_{i\in\mathcal{C}}\mathbb{E}\|w^t - w_{i,K}^{t-1}\|^2} \leq (1 + \beta)\sqrt{\Delta^t}.$$

It could be seen that if we consider the $w_{i,0}^t - w_{i,K}^{t-1}$ as a general update step, it is independent to the dataset. Hence, we assume a virtual update between $w_{i,K}^{t-1}$ and $w_{i,0}^t$ which could be bounded by the divergence term $\Delta^t$. Then we bound the recursive term on $t$.

We know that before $t^\star K + k^\star = t_0$, no different data is sampled, which is, $w_{i,k+1}^t = \widetilde{w}_{i,k+1}^t$ for $\forall\, tK + k \leq t^\star K + k^\star$. After $t_0 + 1$, they become different. Thus, when $t^\star K + k^\star > t_0$, let learning rate $\eta_t$ to be a constant

within each round $t$ and $\eta = \frac{c}{t}$, then we have:

$$\frac{1}{C} \sum_{i \in \mathcal{C}} \mathbb{E}\left[\|w_{i,K}^T - \widetilde{w}_{i,K}^T\| \mid \xi\right] \leq \left(\frac{2L_G}{S} + 2\sigma_l\right) \sum_{t=t^\star K + k^\star}^{TK} \eta_t \exp\left(\left(1 - \frac{1}{S}\right) L \sum_{\tau=t}^{TK} \eta_\tau\right)$$

$$+ (1+\beta)^{\frac{1}{\beta cL}} \sum_{t=t^\star+1}^{T} \exp\left(\left(1 - \frac{1}{S}\right) L \sum_{\tau=t}^{TK} \eta_\tau\right)\sqrt{\Delta^t}.$$

We adopt the same learning rate $\eta = \frac{c}{t}$ where $c = \frac{\mu_0}{K}$ is a positive constant, then

$$\frac{1}{C} \sum_{i \in \mathcal{C}} \mathbb{E}\left[\|w_{i,K}^T - \widetilde{w}_{i,K}^T\| \mid \xi\right]$$

$$\leq 2c\left(\frac{L_G}{S} + \sigma_l\right) \sum_{t=t^\star K + k^\star}^{TK} \frac{1}{t} \exp\left(\left(1 - \frac{1}{S}\right) cL \sum_{\tau=t}^{TK} \frac{1}{\tau}\right)$$

$$+ (1+\beta)^{\frac{1}{\beta cL}} \sum_{t=t^\star+1}^{T} \exp\left(\left(1 - \frac{1}{S}\right) cL \sum_{\tau=t}^{TK} \frac{1}{\tau}\right)\sqrt{\Delta^t}$$

$$\leq 2c\left(\frac{L_G}{S} + \sigma_l\right) \sum_{t=t^\star K + k^\star}^{TK} \frac{1}{t} \exp\left(\left(1 - \frac{1}{S}\right) cL \log\left(\frac{TK}{t}\right)\right)$$

$$+ (1+\beta)^{\frac{1}{\beta cL}} \sum_{t=t^\star+1}^{T} \exp\left(\left(1 - \frac{1}{S}\right) cL \log\left(\frac{TK}{t}\right)\right)\sqrt{\Delta^t}$$

$$\leq 2c\left(\frac{L_G}{S} + \sigma_l\right) (TK)^{\left(1 - \frac{1}{S}\right)cL} \sum_{t=t^\star K + k^\star}^{TK} \left(\frac{1}{t}\right)^{1 + \left(1 - \frac{1}{S}\right)cL} + (1+\beta)^{\frac{1}{\beta cL}} \sum_{t=t^\star+1}^{T} \left(\frac{TK}{t}\right)^{\left(1 - \frac{1}{S}\right)cL}\sqrt{\Delta^t}$$

$$\leq \frac{2\left(L_G + S\sigma_l\right)}{(S-1)L} \left(\frac{TK}{t^\star K + k^\star}\right)^{cL} + (1+\beta)^{\frac{1}{\beta cL}} \sum_{t=t^\star+1}^{T} \left(\frac{TK}{t^\star}\right)^{cL}\sqrt{\Delta^t}.$$

### A.3.3 Proof of Theorem 3

**Theorem 11** *Under the Assumptions 6, 7, 9, and 10, let all conditions above satisfied, we can bound the uniform stability of our proposed FedInit as:*

$$\mathbb{E}\|f(w^{T+1}; z) - f(\widetilde{w}^{T+1}; z)\|$$
$$\leq \frac{U^{\frac{cL}{1+cL}}}{S-1}\left[\frac{2(L_G^2 + SL_G\sigma_l)TK^{cL}}{L}\right]^{\frac{1}{1+cL}} + (1+\beta)^{\frac{1}{\beta cL}}\left[\frac{ULTK}{2(L_G^2 + SL_G\sigma_l)}\right]^{\frac{cL}{1+cL}} \sum_{t=1}^{T}\sqrt{\Delta^t}. \qquad (36)$$

**Proof** *According to Lemma 5, we have:*

$$\mathbb{E}\|f(w^{T+1}; z) - f(\widetilde{w}^{T+1}; z)\| \leq \frac{Ut_0}{S} + \frac{L_G}{C}\sum_{i \in \mathcal{C}}\mathbb{E}\left[\|w_{i,K}^T - \widetilde{w}_{i,K}^T\| \mid \xi\right].$$

*The second term is bounded by uniform stability term as:*

$$\frac{1}{C}\sum_{i \in \mathcal{C}}\mathbb{E}\left[\|w_{i,K}^T - \widetilde{w}_{i,K}^T\| \mid \xi\right] \leq \frac{2\left(L_G + S\sigma_l\right)}{(S-1)L}\left(\frac{TK}{t^\star K + k^\star}\right)^{cL} + (1+\beta)^{\frac{1}{\beta cL}}\sum_{t=t^\star+1}^{T}\left(\frac{TK}{t^\star}\right)^{cL}\sqrt{\Delta^t}$$

$$\leq \frac{2\left(L_G + S\sigma_l\right)}{(S-1)L}\left(\frac{TK}{t^\star K + k^\star}\right)^{cL} + (1+\beta)^{\frac{1}{\beta cL}}\sum_{t=1}^{T}\left(\frac{TK}{t^\star}\right)^{cL}\sqrt{\Delta^t}.$$

*Let the $t_0 = t^\star K + K^\star = \left[\frac{2(L_G^2 + SL_G\sigma_l)}{UL}(TK)^{cL}\right]^{\frac{1}{1+cL}}$, then $t^\star > \left[\frac{2(L_G^2 + SL_G\sigma_l)}{UL}\right]^{\frac{1}{1+cL}}\frac{T^{\frac{cL}{1+cL}}}{K^{\frac{1}{1+cL}}}$ we have:*

$$\mathbb{E}\|f(w^{T+1}; z) - f(\widetilde{w}^{T+1}; z)\|$$

$$\leq \frac{U^{\frac{cL}{1+cL}}}{S-1}\left[\frac{2(L_G^2 + SL_G\sigma_l)}{L}\right]^{\frac{1}{1+cL}}(TK)^{\frac{cL}{1+cL}}$$

$$+ (1+\beta)^{\frac{1}{\beta cL}}\left[\frac{UL}{2(L_G^2 + SL_G\sigma_l)}\right]^{\frac{cL}{1+cL}}(TK)^{\frac{cL}{1+cL}}\sum_{t=1}^{T}\frac{\sqrt{\Delta^t}}{T}$$

$$= \frac{U^{\frac{cL}{1+cL}}}{S-1} \left[ \frac{2(L_G^2 + SL_G\sigma_l)(TK)^{cL}}{L} \right]^{\frac{1}{1+cL}} + (1+\beta)^{\frac{1}{\beta cL}} \left[ \frac{ULTK}{2(L_G^2 + SL_G\sigma_l)} \right]^{\frac{cL}{1+cL}} \sum_{t=1}^{T} \frac{\sqrt{\Delta^t}}{T}.$$

*This completes this proof.*

# B  Experiments

In this section, we mainly provide the detailed experimental setups in our paper, including the introduction of the benchmarks, dataset, hyperparameters selections, and adding some more experiments.

## B.1  Setups

**Dataset.**   We follow the previous works and select the CIFAR-10/100 [20] dataset in our experiments. In the CIFAR-10 dataset, there is a total of 50,000 training images and 10,000 test images which contain 10 categories. Each data sample is a color image with a size of $32 \times 32$. In the CIFAR-100 dataset, there is also a total of 50,000 training images and 10,000 test images. It contains 100 categories of the same size as CIFAR-10. For their limited resolutions, we only use general data augmentations. On each local heterogeneous dataset, we use general normalization on the images with specific mean and variance. For the training process, we randomly crop a $32 \times 32$ patch from the vanilla images with a zero padding of $4$. For the test process, we use the raw images.

**Heterogeneity.**   We follow Hsu et al. [14] to introduce the label imbalance as the heterogeneous dataset. According to the Dirichlet distribution, we first generate a specific vector with respect to a constant $D_r$ to control its variance level. Usually, heterogeneity becomes stronger when $D_r$ decreases. Then according to the vector, we sample the images from the training dataset. Here we enable the sampling with replacement to generate the local dataset, which means the local clients may have the same data sample if they are assigned to the same category. This is more related to the real scenario. At the same time, it also will lose some data samples, we assume this case is due to the offline devices. This is a common case because the FL has an unreliable network connection across the devices.

**Benchmarks.**   In this paper, we use *FedAvg* [26], *FedAdam* [30], *FedSAM* [29], *SCAFFOLD* [19], *FedDyn* [1], and *FedCM* [46] as the benchmarks. *FedAvg* propose the general FL paradigm based on the local SGD method. It allows partial participation training via uniformly selecting a subset of local clients. A series of developments followed it to improve its performance. *FedAdam* studies the efficient adaptive optimizer on the global server update, which extends the scope of the FL paradigm. *SCAFFOLD* indicates that FL suffers from the client-drift problem which is due to the inconsistency of local optimum. Beyond this, it uses the variance reduction technique to further reduce the divergence across the local clients. To further alleviate, *FedDyn* studies the primal-dual method via adopting the ADMM to solve the problem. The consistency condition works as a constraint during the optimization. It proves that when the global model converges, the local objectives will be aligned with the global one. *FedCM* proposes an efficient momentum-based method, dubbed client-level momentum. It communicates the global update as a correction to correct each local update to force the local client updates in a similar direction. It maintains very high consistency via a biased correction. Therefore, it relies on an accurate global direction estimation. *FedSAM* considers the generalization performance. Generally, we adopt empirical risk minimization (ERM) to perform the optimization process. While the sharpness-aware-minmization (SAM) studies that it could search for a flat loss landscape. Flatness guarantees a higher generalization performance. Though our focus is not the generalization, we theoretically prove that even in the *FedAvg* method divergence term affects the generalization error bound more than the optimization error bound. From this perspective, generalization-efficiency methods may also be connected with consistency guarantees. These are all the SOTA benchmarks in the FL community that concern more on enhancing consistency.

**Hyperparameters selection.**   Here we detail our hyperparameter selection in our experiments. For each splitting, we fix the total communication rounds $T$, local interval $K$, and mini-batchsize for all the benchmarks and our proposed *FedInit*. The other selections are stated as follows.

$\star$ means different selections according to the specific setups.

We fix the most hyperparameters of testing the whole benchmarks for a fair comparison. The other algorithm-specific hyperparameters are subjected to specific circumstances. The ResNet-18-GN and VGG-11 adopt the same set of selections. Then we show algorithm-specific hyperparameters:

**Special hyperparameter selections**. In the *FedAdam* method, we test that it is very sensitive to the global learning rate. Though we report the best selection is $0.1$, it still requires some finetuning based on the dataset and experimental setups. In the *FedSAM* method, we test it is very sensitive to the perturbation learning rate. Usually, it should be selected as $0.1$ in most cases. However, in some poor-sampling cases, i.e. low participation ratio, it should be selected as $0.01$. In the *FedDyn*, we test it is very sensitive to the regularization coefficient.

Table 6: General hyperparameters introductions.

| Dataset | CIFAR-10 | best selection |
|---|---|---|
| communication round $T$ | 500 | - |
| local interval $K$ | 5 | - |
| minibatch | 50 | - |
| weight decay | $1e^{-3}$ | - |
| local learning rate | $[0.01, 0.1, 0.5, 1]$ | 0.1 |
| global learning rate | $[0.01, 0.1, 1.0]$ | 1.0/0.1 |
| learning rate decay | $[0.995, 0.998, 0.9995]$ | 0.998 |
| relaxed coefficient $\beta$ | $[0.01, 0.02, 0.05, 0.1, 0.15]$ | 0.1/0.01 |
| Dataset | CIFAR-100 | best selection |
| communication round $T$ | 500 | - |
| local interval $K$ | 5 | - |
| minibatch | 50 | - |
| weight decay | $1e^{-3}$ | - |
| local learning rate | $[0.01, 0.1, 0.5, 1]$ | 0.1 |
| global learning rate | $[0.01, 0.1, 1.0]$ | 1.0/0.1 |
| learning rate decay | $[0.998, 0.9995, 0.9998]$ | 0.998/0.9995 |
| relaxed coefficient $\beta$ | $[0.01, 0.02, 0.05, 0.1, 0.15]$ | $\star$ |

Table 7: Algorithm-specific hyperparameter introductions.

| Method | specific hyperparameter | introduction | selection | best selection |
|---|---|---|---|---|
| FedAdam | global learning rate | adaptive learning rate | $[0.01, 0.05, 0.1, 1]$ | 0.1 |
| FedSAM | perturbation learning rate | ascent step update | $[0.01, 0.1, 1]$ | 0.1 |
| FedDyn | regularization coefficient | coefficient of prox-term | $[0.001, 0.01, 0.1, 1]$ | $\star$ |
| FedCM | client-level coefficient | ratios in local updates | $[0.05, 0.1, 0.5, 0.9]$ | $\star$ |

Generally, it adopts the regularization coefficient to be 0.1 on CIFAR-10 and 0.01/0.001 on CIFAR-100. In *FedCM*, we select the client-level coefficient as 0.1 which is followed by Xu et al. [46] in most cases. However, on the VGG-11 model, it fails to converge with a small client-level coefficient.

## B.2 Experiments

### B.2.1 Curves

In this section, we show the curves of our results.

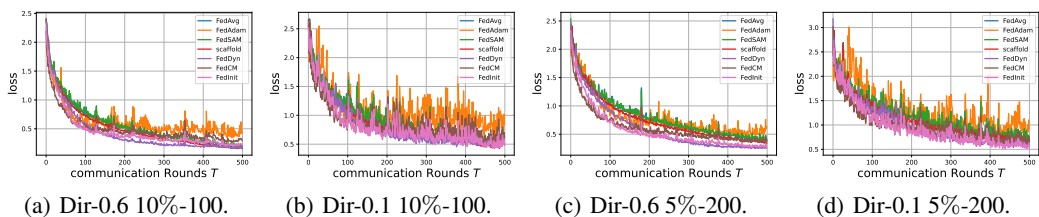

(a) Dir-0.6 10%-100.    (b) Dir-0.1 10%-100.    (c) Dir-0.6 5%-200.    (d) Dir-0.1 5%-200.

Figure 3: Loss on the CIFAR-10 dataset.

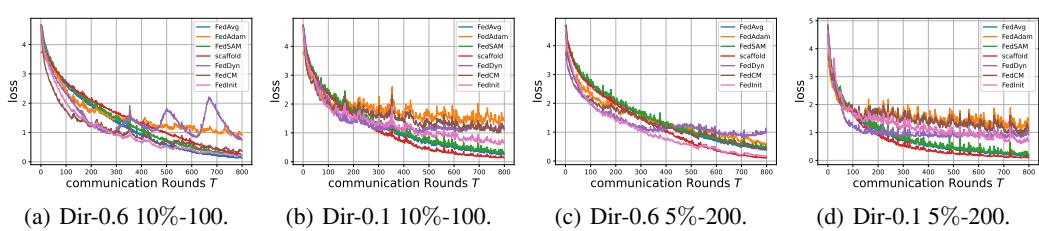

(a) Dir-0.6 10%-100.    (b) Dir-0.1 10%-100.    (c) Dir-0.6 5%-200.    (d) Dir-0.1 5%-200.

Figure 4: Loss on the CIFAR-100 dataset.

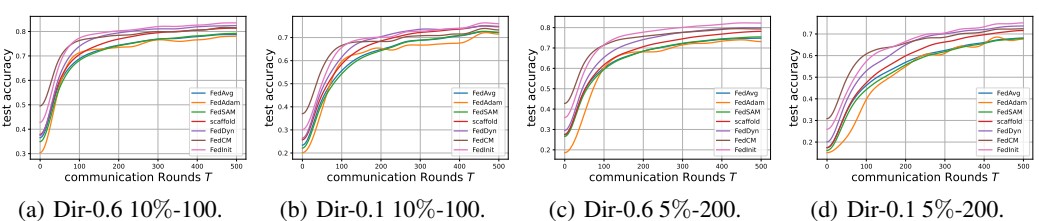

(a) Dir-0.6 10%-100.    (b) Dir-0.1 10%-100.    (c) Dir-0.6 5%-200.    (d) Dir-0.1 5%-200.

Figure 5: Test accuracy on the CIFAR-10 dataset.

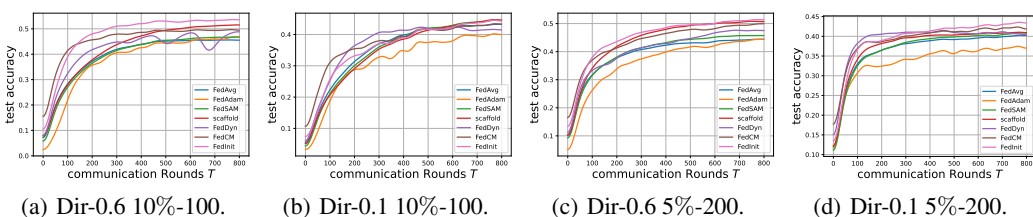

(a) Dir-0.6 10%-100.    (b) Dir-0.1 10%-100.    (c) Dir-0.6 5%-200.    (d) Dir-0.1 5%-200.

Figure 6: Test accuracy on the CIFAR-100 dataset.

To show the stable accuracy curves, we use the third-party `tsmoothie.smoother` to smooth the raw curve via the function `ConvolutionSmoother(window_len=100, window_type='hanning')`. On most setups, our proposed *FedInit* achieves the SOTA results. It effectively avoids negative impacts from local overfitting.

### B.2.2 Consistency of Different Initialization

In this part, we mainly test the consistency level of different $\beta$. The coefficient $\beta$ controls the divergence level of the local initialization states. We select the *FedAvg* and *SCAFFOLD* to show the efficiency of the proposed relaxed initialization.

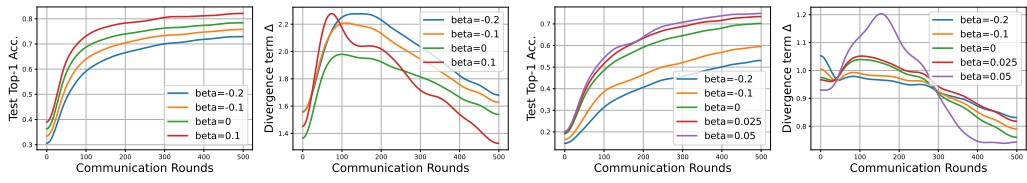

(a) Dir 0.6 10%-100 Accuracy and Consistency.  (b) Dir 0.1 5%-200 Accuracy and Consistency.

Figure 7: Experiments of *FedAvg* on the CIFAR-10 dataset.

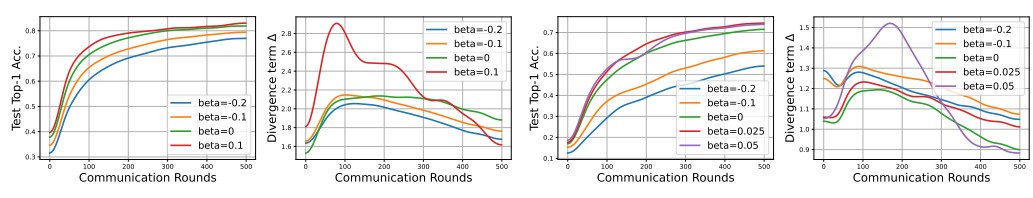

(a) Dir 0.6 10%-100 Accuracy and Consistency.  (b) Dir 0.1 5%-200 Accuracy and Consistency.

Figure 8: Experiments of *SCAFFOLD* on the CIFAR-10 dataset.

These experiments show that the relaxed initialization (RI) effectively reduces the consistency and improves the test accuracy. In all tests, when $\beta = 0$ (green curve), it represents the vanilla method without RI. After incorporating the RI, the test accuracy achieves at least 2% improvement on each setup.

### B.2.3 Communication, Calculation and Storage Costs

In this part, we mainly compare the communication, calculation, and storage costs theoretically and experimentally. By assuming the total model maintain $d$ dimensions, we summarize the costs of benchmarks and our proposed *FedInit* as follows:

Table 8: Communication, calculation, and storage costs per communication round.

| Method | communication | ratio | gradient calculation | ratio | total storage | ratio |
|--------|:---:|:---:|:---:|:---:|:---:|:---:|
| FedAvg | $Nd$ | $1\times$ | $NKd$ | $1\times$ | $Cd$ | $1\times$ |
| FedAdam | $Nd$ | $1\times$ | $NKd$ | $1\times$ | $Cd$ | $1\times$ |
| FedSAM | $Nd$ | $1\times$ | $2NKd$ | $2\times$ | $2Cd$ | $2\times$ |
| SCAFFOLD | $2Nd$ | $2\times$ | $NKd$ | $1\times$ | $3Cd$ | $3\times$ |
| FedDyn | $Nd$ | $1\times$ | $NKd$ | $1\times$ | $3Cd$ | $3\times$ |
| FedCM | $2Nd$ | $2\times$ | $NKd$ | $1\times$ | $2Cd$ | $3\times$ |
| FedInit | $Nd$ | $1\times$ | $NKd$ | $1\times$ | $Cd$ | $1\times$ |

where $N$ is the number of participating clients, $C$ is the total number of clients, and $K$ is the local training interval.

**Limitations of the benchmarks.** From this table, we can see that *SCAFFOLD* and *FedCM* both require double communication costs than the vanilla *FedAvg*. They adopt the correction term (variance reduction and client-level momentum) to revise each local iteration. Though this achieves good performance, we must indicate that under the millions of edge devices in the FL paradigm, this may introduce a very heavy communication bottleneck. In addition, the *FedSAM* method considers adopting the local SAM optimizer instead of ERM to approach the flat minimal. However, it requires double gradient calculations per iteration. For the very large model, it brings a large calculation cost that can not be neglected. *SCAFFOLD* and *FedDyn* are required to store $3\times$ vectors on each local devices. This is also a limitation for the light device, i.e. mobiles.

We also test the practical wall-clock time on real devices. Our experiment environments are stated as follows:

Table 9: Experiment environments.

| GPU | CUDA | Driver Version | CUDA Version | Platform |
|:---:|:---:|:---:|:---:|:---:|
| Tesla-V100 (16GB) | NVIDIA-SMI 470.57.02 | 470.57.02 | 11.4 | Pytorch-1.12.1 |

In the following table, we test the wall-clock time cost of each method:

Table 10: Wall-clock time cost (s/round).

| | FedAvg | FedAdam | FedSAM | SCAFFOLD | FedDyn | FedCM | FedInit |
|--------|:---:|:---:|:---:|:---:|:---:|:---:|:---:|
| 10%-100 | 19.38 | 23.22 | 30.23 | 28.61 | 23.84 | 22.63 | **20.41** |
| ratio | $1\times$ | $1.19\times$ | $1.56\times$ | $1.47\times$ | $1.23\times$ | $1.17\times$ | **$1.05\times$** |
| 5%-200 | 15.87 | 17.50 | 22.18 | 24.49 | 20.61 | 18.19 | **16.14** |
| ratio | $1\times$ | $1.10\times$ | $1.40\times$ | $1.54\times$ | $1.30\times$ | $1.15\times$ | **$1.02\times$** |

From this table, due to the different communication costs and calculation costs, the practical wall-clock time is different for each method. Generally, *FedAvg* adopts the local-SGD updates without any additional calculations. *FedAdam* adopts similar local-SGD updates and an adaptive optimizer on the global server. *FedSAM* calculation double gradients, which is the main reason for being slowest among the benchmarks. *SCAFFOLD*, *FedDyn*, and *FedCM* are required to calculate some additional vectors to correct the local updates. Therefore they need some additional time costs. Our proposed *FedInit* only adopts an additional initialization calculation, which requires the same costs as *FedAvg*.

### B.2.4 Training Efficiency: Communication Rounds and Time Costs

In this part, we mainly show the results of the training efficiency. We set the target accuracy and compare their required communication rounds and training time respectively. We test on the ResNet-18-GN model with the 10%-100 Dir-0.1 splitting.

Table 11: We train 500 rounds on CIFAR-10 and 800 rounds on CIFAR-100. "-" means the corresponding method can not achieve the target accuracy during the training processes.

| Method | CIFAR-10 ($\geq$70%) | | | | CIFAR-100 ($\geq$30%) | | | |
|---|---|---|---|---|---|---|---|---|
| | Round | | Time (s) | | Round | | Time (s) | |
| | | Speed Ratio | | Speed Ratio | | Speed Ratio | | Speed Ratio |
| FedAvg | 371 | 1× | 7189 | 1× | 191 | 1× | 3701 | 1× |
| FedAdam | 489 | 0.76× | 11354 | 0.63× | 256 | 0.74× | 5944 | 0.62× |
| FedSAM | 377 | 0.98× | 11396 | 0.63× | 204 | 0.93× | 6166 | 0.60× |
| SCAFFOLD | 248 | 1.50× | 7095 | 1.01× | 211 | 0.90× | 6036 | 0.61× |
| FedDyn | 192 | 1.93× | 4577 | 1.57× | 122 | 1.56× | 2908 | 1.27× |
| FedCM | 183 | 2.02× | 4141 | 1.73× | **95** | **2.01×** | **2149** | **1.72×** |
| **FedInit** | **172** | **2.15×** | **3510** | **2.04×** | 132 | 1.44× | 2694 | 1.37× |

The setups of the test environment are stated in Table 9. According to this table, we clearly see that some advanced methods, i.e. *SCAFFOLD* and *FedDyn*, are efficient on the communication round $T$. However, due to the additional costs of each training iteration, they must spend more time on the total training. *FedInit* is a very light and practical method, which only adopts a relaxed initialization on the *FedAvg* method, which makes it to be better and even achieves SOTA results.

