# OpenReview forum: "Understanding How Consistency Works in Federated Learning via Stage-wise Relaxed Initialization"
_NeurIPS.cc/2023/Conference — NeurIPS 2023 poster_

### Official Review · Reviewer_g4w5 · 2023-06-29

**Soundness:** 2 fair
**Presentation:** 2 fair
**Contribution:** 2 fair
**Rating:** 5
**Confidence:** 3

**Summary:**

The manuscript proposes a novel federated learning method to alleviate the negative impact of the "client drift" problem and enhance consistency in the FL paradigm. The manuscript also analyzes the intrinsic impact of local consistency on optimization error, test error, and generalization error. Several experiments are conducted to validate the efficiency of the proposed method.

**Strengths:**

1. The proposed method outperforms baseline methods without increasing communication costs and can be easily incorporated into other methods.
2. The manuscript provides ample theoretical analysis to demonstrate the theoretical bounds of the proposed method.

**Weaknesses:**

1. The hyperparameters were not tuned for different methods. Since the experiments are highly sensitive to hyperparameter settings, the superiority of the proposed method is likely due to the selection of hyperparameters. Given that the personalized process proposed in the manuscript bears resemblance to the momentum of parameter updates, I suspect that when tuning the learning rate for the baseline method, the baseline method may outperform the proposed method.
2.  The results shown in Figure 1 suggest that the proposed method is highly sensitive to the choice of beta.

**Questions:**

1. The manuscript thinks that inconsistent local optima cause "client drift," which results in the inferiority of federated learning (FL) algorithms. The proposed method aims to alleviate this divergence by employing personalized relaxed initialization at the beginning of each round. I have a question: based on my understanding, personalized initialization in each round would lead to greater differences across different clients. So, how does this strategy help alleviate the divergence?
2. After a brief review of the theoretical analysis, I am uncertain if I correctly understood the proof. In formulas (9) and (14), the upper bounds achieve their minimum when beta=0. If that's the case, why does the personalized relaxation contribute to improving the performance of FL algorithms?

**Limitations:**

The experiments in the manuscript are not rigorous enough as the hyperparameters for different methods were not tuned. This could potentially lead to an overestimation of the performance of the proposed method.

---

> ### Author Rebuttal · Authors · 2023-08-07
>
> Thank you very much for your review and comments on our work. We'll answer your questions one by one, including some misunderstandings and some essential academic questions. We are also very honored to share some of our understandings with you.
>
>
> ## About the question of "The hyperparameters were not tuned for different methods.":
>
> **We have finetuned the hyperparameters and all baselines are carefully tuned and compared fairly.** Due to the page limitation, the details are stated on **Line.738 (Table.6)** and **Line.742 (Table.7)** in the **Appendix** and we also submit the code in the Supplementary Material.
>
> We test all algorithms and select parameters that most algorithms can be trained well to report as the best selection in Table.6, which is also widely used in many previous works. For instance, in [1], they provide the hyperparameters on the page.15 (Appendix) which are similar to ours. In [2], they provide the hyperparameters in Section E.2, which adopt the same learning rate as ours. In [3], they provide the hyperparameters selection in section C.2 and most parameters are the same as ours. Additionally, we also finetuned personalized hyperparameters in Table.7. Moreover, we use the same set of hyperparameters to test the experiments of adding the relaxed initialization as a plug-in to the current methods, i.e. SCAFFOLD and FedSAM, and they also could be improved by the RI, as shown in Table.2. Our proposed RI is not a trick based on the specific hyperparameters selection. It works both theoretically and experimentally. Additional experiments of re-tuning lr are still running and we will reply soon when it is finished.
>
> **We also use the sampling with replacement in the dataset split**. Compared to **sampling without replacement**, our split is more difficult and the performance will drop a lot on all methods. It means one data sample may exist in more local clients. This situation is more realistic because, on large-scale federated clients, the probability of data duplication is very high.
>
> [1] Federated Learning Based on Dynamic Regularization (ICLR 2021)
>
> [2] Generalized Federated Learning via Sharpness Aware Minimization (ICML 2022)
>
> [3] FedCM: Federated Learning with Client-level Momentum
>
>
> ## About the question "The results shown in Figure 1 suggest that the proposed method is highly sensitive to the choice of beta.":
>
> We have proven that the coefficient $\beta$ has a maximum value (see $\beta$ selection range in Theorem 6). If it exceeds this maximum value, the method may not converge. Figure.1 demonstrates that our proof matches the experiments. In the experiments, we suggest selecting this coefficient as a small positive value, i.e. 0.05, 0.1. It could be seen in Figure.1, it is very stable in its valid range.
>
>
> ## About the question "I have a question: based on my understanding, personalized initialization would lead to greater differences across different clients. So, how does this strategy help alleviate the divergence?":
>
> Our motivation is that if the current initialization is farther away from the local optimal solution than the global server model, then after the local training is finished it will also be farther away from the local optimal solution than before. To implement this, we propose relaxed initialization (RI). The working mode of RI is similar to the idea of "​​lookahead". Differently, (1) "​​lookahead" only works at the end of each stage; (2) "​​lookahead" only works for the global models on the global server. However, RI helps each local client to backtrack a small distance at the beginning of each stage. Therefore, after the local training in the next stage, the trained local model will get close to each other than before. We provide a figure to illustrate the principles of the relaxed initialization in the one-page .pdf file.
>
>
> ## About the question of "In formulas (9) and (14), the upper bounds achieve their minimum when beta=0.":
> Formulas (9) and (14) are not the final conclusion. We aim to find the best selection for the excess risk instead of the optimization only. Here we briefly introduce the final conclusion in section 4.3.4 (formula (18)).
>
> We combine the Theorem.2, 3, 5 and bound the excess risk as:
> $$
> \mathcal{E}_E\leq \widetilde{\mathcal{O}}\left(\frac{D+L(\sigma_l^2 + KG^2)}{NKT}\right) + \mathcal{O}\left(\frac{1}{S}\left[\sigma_l(TK)^{cL}\right]^\frac{1}{1+cL}\right) + \widetilde{\mathcal{O}}\left(\frac{\sqrt{D+G^2}K^\frac{cL}{1+cL}}{T^\frac{1}{1+cL}}\right).
> $$
>
> Formula (9) only indicates the optimization bias term. Formula (14) indicates the latter two terms as the generalization bias. Because the constant part in the convergence term $\Delta^t$ also contains $\beta$, formula (14) is not the final version. In above upper bound, it contains three main parts. The first part is the optimization error, the second part is the stability bias which is related to the total number of samples, and the dominant part is the divergence bias which comes from the consistency term. Here we omit the constant part on the above formula. **In the paragraph "Selection of $\beta$" (Line.284)**, we discuss it in detail. The constant part of the dominant term (related to $\beta$) is $\frac{(1+\beta)^{\frac{1}{\beta cL}}}{\sqrt{1-96\beta^2}}$ (Line.287) where $0<\beta<\frac{\sqrt{6}}{24}$. It can be easily verified that when $\beta$ is selected as a specific positive number the constant term achieves the minimal.
>
> We summarize them as follows:
>
> (1) $\beta=0$ only helps to minimize the optimization error.
>
> (2) It is a trade-off between optimization and excess risk.
>
> (3) To minimize the dominant term of the excess risk, $\beta$ should be selected as a small positive number instead of 0.
>
> ### It is a pleasure to discuss this with you, which will help us to improve this work further. We explain and prove the concerns mentioned in the reviews. If there are any questions, we are happy to continue the discussion with you. Thank you again for reading this rebuttal.

---

> > ### Author Response · Authors · 2023-08-11
> > **Additional Experiments on Tuning Learning Rate**
> >
> > Dear reviewer g4w5, to address your concerns on the finetuning learning rates, we re-tuned the learning rate to show that our hyperparameters selection is correct. Due to the rebuttal time limitation, we test the experiments on the following setup.
> >
> > CIFAR-10 Dirichlet-0.1 split, 10 active clients of total 100 clients, other hyperparameters are selected as we introduced in our paper (Appendix)
> >
> > Generally, the global learning rate is set as 1.0 in FL except for some specific methods. Global lr=1.0 means to average the local models. We follow this to fix the global learning rate and finetune the local learning rate as follows.
> >
> > |  Method   |  lr=0.05  |  lr=0.1  |  lr=0.2  |   lr=0.5  |
> > |  :----:  |  :----:   | :----:  |  :----:   | :----:   |
> > | SCAFFOLD  | 73.65% | **75.19%** | 74.93% | 53.16% |
> > | FedCM  | 69.56% | **74.14%** | 73.73% | 60.32% |
> > | FedDyn  | 71.74% |  75.10%  | **75.27%**  |   -   |
> >
> > "-": can not converge
> >
> > As shown in our paper (Appendix), the best selection of the local learning rate is 0.1, which could fairly compare the performance of the whole baselines. We finetune the hyperparameters in a wide range and report the fair and best selection. We also learn from the classical paper [1,2,3] to finish the experiments. Moreover, relaxed initialization could further improve the performance of some current advanced methods. Both theoretical analysis and experiments verify its efficiency. **Relaxed initialization is not a trick that relies on finetuning the learning rate.**
> >
> > [1] Federated Learning Based on Dynamic Regularization (ICLR 2021)
> >
> > [2] Generalized Federated Learning via Sharpness Aware Minimization (ICML 2022)
> >
> > [3] FedCM: Federated Learning with Client-level Momentum

---

> > > ### Comment · Reviewer_g4w5 · 2023-08-11
> > > **About hyper-parameter**
> > >
> > > Thanks for the response. Could you please add some experiments with both different local intervals and learning rates for baseline methods? Thanks.

---

> > > > ### Author Response · Authors · 2023-08-11
> > > > **Re: About hyper-parameter**
> > > >
> > > > Thank you very much for your quick reply.
> > > >
> > > > We currently use 5 local epochs in our experimental setups. We will test the experiments on 10 local epochs. The search range of the local learning rate is between 0.01 and 0.5 and the global learning rate will still be fixed as 1.0. We will report the results when they are finished ASAP. Thanks!

---

> > > > ### Author Response · Authors · 2023-08-14
> > > > **Additional Experiments on Tuning Learning Rate on Different Local Epochs**
> > > >
> > > > Dear reviewer g4w5, we extend the local epochs to 10 and tune the learning rate for the baselines. Results are shown in the following table.
> > > >
> > > > CIFAR-10 Dirichlet-0.1 split, 10 active clients of total 100 clients, local epochs=10, other hyperparameters are selected as we introduced in our paper (Appendix)
> > > >
> > > > |    Method   |   lr=0.01  |   lr=0.05  |   lr=0.1  |   lr=0.2  |   lr=0.5  |
> > > > |   :----:    |   :----:   |  :----:   |  :----:   |  :----:   |  :----:   |
> > > > | SCAFFOLD  | 67.47% |  74.34%  | **75.74%** |  75.01%  |  49.03%  |
> > > > | FedCM         | 74.06% |  **74.94%**  |  74.23%  |  68.60% |  59.26% |
> > > > | FedDyn        | 69.37% |  73.05%  |  **75.29%**  |   74.10%  |  -  |
> > > > | FedInit         |  62.82% |  73.50% |  **75.48%**  |   73.82%  |  63.43%  |
> > > >
> > > > "-": can not converge
> > > >
> > > > When local epochs are extended to 10, the results are similar to the selection of 5. As claimed in [1], local clients could be optimized well enough for FL by adopting 5 local epochs. Similar experiments also could be seen in [2] (figure 2 in their paper). FedInit still could achieve comparable performance with the current advanced methods. **And different from the baselines, FedInit only communicates one vector and does not require any additional communication costs, which makes it very useful in practical scenarios.**
> > > >
> > > > Moreover, relaxed initialization still could help the advanced methods to improve their performance. As shown in the following Table, hyperparameters are selected as the best group above,
> > > >
> > > > |    Method   |   vanilla  |   + FedInit  |
> > > > |   :----:    |   :----:   |  :----:   |
> > > > | SCAFFOLD  | 75.74% |  **76.34%**  |
> > > > | FedCM         | 74.94% |  **75.10%**  |
> > > > | FedDyn        | 75.29%  |   **75.42%**  |
> > > >
> > > > The coefficient of relaxed initialization (RI) for SCAFFOLD is selected as 0.1, and 0.02 for the others.
> > > >
> > > > As we claimed before, the theoretical analysis indicates that RI benefits from a positive coefficient instead of 0 in practical training, which could minimize the excess risk in the FL paradigm. Experimental results also validate its efficiency.
> > > >
> > > > [1] Federated Learning Based on Dynamic Regularization (ICLR 2021)
> > > >
> > > > [2] Generalized Federated Learning via Sharpness Aware Minimization (ICML 2022)

---

> > > > > ### Comment · Reviewer_g4w5 · 2023-08-14
> > > > >
> > > > > Thanks for the response.
> > > > >
> > > > > The updated results correspond with those presented in Table 1 and Table 2 of the original manuscript, correct?
> > > > >
> > > > > In Table 1 of the original manuscript, for the cifar-10 dataset, under the conditions of 10%-100 clients and Dir-0.1, the optimal performance achieved by SCAFFOLD is reported as 75.06±0.16, while the optimal performance achieved by FedInit is reported as 75.95±0.19. However, in the revised table provided by the author, following hyper-parameter fine-tuning (local epochs=10 and learning rate=0.1), the best performance of SCAFFOLD is reported as 75.74%, which is quite comparable to FedInit's best performance presented in Table 1. As such, I am inclined to disagree with the author's assertion that "the results are similar to the selection of 5". It is plausible that the performance of baseline methods could be further enhanced by extending the number of local epochs (perhaps the author could perform experiments with SCAFFOLD alone for efficiency). If this hypothesis holds true, particularly for the outcomes showcased in Table 1, the claimed superiority of FedInit may not be as compelling.
> > > > >
> > > > > Actually, the proposed method is likely to enhance model performance by leveraging an increased volume of updates in each round. Consequently, when the baseline methods fall short of full convergence within each round, it is reasonable to expect that the proposed approach will outperform them.

---

> > > > > > ### Author Response · Authors · 2023-08-14
> > > > > > **Additional Experiments on Local Epochs=15 and Rebuttal**
> > > > > >
> > > > > > Dear reviewer g4w5, thanks for your quick reply. We finish the experiments on the local epochs=15 as shown in the following.
> > > > > >
> > > > > > ### About the question of "It is plausible that the performance of baseline methods could be further enhanced by extending the number of local epochs (perhaps the author could perform experiments with SCAFFOLD alone for efficiency). If this hypothesis holds true, particularly for the outcomes showcased in Table 1, the claimed superiority of FedInit may not be as compelling."
> > > > > >
> > > > > > **This hypothesis is not correct.** If we extend the local epochs to a very large number, the local client will be optimized well and almost achieve the local minimal. This is fatal for federated learning. All local models will overfit the local dataset and are far away from the global optimal. This is the "client-drift" problem proposed in [1], which could be formulated as $\frac{1}{m}\sum_i x_i^\star \neq x^\star$. This is also one of the major issues in FL. Therefore, the local epochs can not be extended too much. As shown in the following table, under large local epochs, SCAFFOLD will also be affected.
> > > > > >
> > > > > > |  Method  |  E=1  |  E=5  |  E=10  |  E=15  |
> > > > > > |  :----:  | :----:| :----:| :----: | :----: |
> > > > > > | SCAFFOLD | 71.51% | 75.06%| **75.74%** | 75.25%|
> > > > > >
> > > > > > Actually, many previous works [1,2,3] have been devoted to preventing local clients from fully converging. [1] uses the variance reduction to correct the local gradient as the global gradient estimation to avoid overfitting. [2] changes the local objective from $F_i(w_i)$ to $F_i(w_i)+\lambda\Vert w_i-w\Vert^2$ to force the local model to be close to the last global state. [3] uses the ADMM method to change the local objective. However, even with these advanced methods, the local interval still cannot be set too large. Training with large intervals will eventually degrade performance and fall into the average of the local minimal $\frac{1}{m}\sum_i x_i^\star$ instead of the global minimal $x^\star$. Our proposed FedInit also could work as a light plug-in and help them to further improve the performance without conflicts.
> > > > > >
> > > > > >
> > > > > > [1] SCAFFOLD: Stochastic Controlled Averaging for Federated Learning
> > > > > >
> > > > > > [2] Federated Optimization in Heterogeneous Networks
> > > > > >
> > > > > > [3] Federated Learning Based on Dynamic Regularization
> > > > > >
> > > > > >
> > > > > > ### About the fair comparison:
> > > > > >
> > > > > > As we mentioned above, due to the "client-drift" problem in FL, increasing the local interval does not always improve the results of the algorithm. So about your question about SCAFFOLD outperforming FedInit on local epochs=10, that is not fair. Here we make a comprehensive table comparing the performance of SCAFFOLD and FedInit and their training costs.
> > > > > >
> > > > > > Under the setup in our paper, 1 epochs=10 iterations. After 500 communication rounds, the total iterations is 5000 if local epoch=1.
> > > > > >
> > > > > > |          |  E=1  |  E=5  |  E=10  |  E=15  |
> > > > > > |  :----:  | :----:| :----:| :----: | :----: |
> > > > > > | SCAFFOLD | 71.51% | 75.06%| **75.74%** | 75.25%  |
> > > > > > | FedInit  | 73.11% | **75.95%** | 75.48% | 73.92% |
> > > > > > | Total Iterations  | 5000 | 25000| 50000 | 75000  |
> > > > > >
> > > > > > As stated above, **to achieve the best results, FedInit requires fewer iterations and less training time**. Furthermore, SCAFFOLD requires to communicate 2 vectors per communication round and FedInit only requires to communicate 1 vector. Another main issue in FL is the communication bottleneck in the practical training. Therefore, the proposed FedInit is better for its fewer communication costs and running wall-clock time costs. SCAFFOLD applies a variance reduction technique, which does not conflict with our relaxed initialization. What we would like to claim is that SCAFFOLD adopts the variance reduction and double communication costs to improve performance, while FedInit can achieve comparable performance with fewer training costs.

---

> > > > > > > ### Comment · Reviewer_g4w5 · 2023-08-15
> > > > > > >
> > > > > > > Thanks for all the explanation. I have updated my score to 5.

---

> > > > > > > > ### Author Response · Authors · 2023-08-15
> > > > > > > > **Thank you very much for this discussion**
> > > > > > > >
> > > > > > > > Dear reviewer g4w5, thank you very much for your affirmation of our work. The discussion also helps us to further improve this work. We will fix all the typos and misunderstandings above in the next version. Experiments will be added in the appendix due to the page limitation. Thank you again for this detailed discussion!

---

### Official Review · Reviewer_sALG · 2023-06-30

**Soundness:** 3 good
**Presentation:** 3 good
**Contribution:** 3 good
**Rating:** 6
**Confidence:** 3

**Summary:**

This paper proposes to initialize the local state by moving away from the current global state toward the reverse direction of the latest local state. They demonstrate theoretically and empirically that this revision can help consistency for better performance. The method is also a practical plug-in that could easily to incorporated into other methods.

**Strengths:**

- This paper goes deeper into how consistency work in FL systems, and provides a simple and effective initialization-based solution.
- This paper gives a comprehensive theoretical analysis of the problem, which I think is a good contribution.
- The paper is well-written and easy to follow.
- Experiment shows good performance.

**Weaknesses:**

- I understand there might not be enough space for experiments and the focus looks to be theoretical analysis, but it would be better if the authors can give more experimental results for different FL settings, e.g., more clients/datasets, which may help evaluate the method better. For example,  100 clients/10% participation rate and 200 clients/ 5% participation rate can be relatively limited. Maybe can we have a curve for participation rate with the same amount of clients and perhaps a curve for number of clients to see how these FL settings influence the performance?

**Questions:**

- Does the method also work well for a large number of clients, e.g., 1000?
- How does participation rate influence performance?

**Limitations:**

The authors have stated the limitation and future work as pFL.

---

> ### Author Rebuttal · Authors · 2023-08-07
>
> Thank you very much for your review and affirmation of our work. We'll answer your questions one by one in the following, including some misunderstandings and some essential academic questions worth exploring. We are also very honored to share some of our understandings with you.
>
>
> ## About the question "Does the method also work well for a large number of clients, e.g., 1000?"
>
> Thank you very much for this question. Large-scale training is one of the goals pursued by federated learning. We finish the experiments on a total of 1000 clients and summarize their results in the following table.
>
> #### Total clients=1000, CIFAR-10, Partial-participation=0.01 (10 active clients/round), Communication round=500, Dirichlet-0.1 split
>
> Due to the very large number of clients, each client is provided with 50 training samples only. In order to avoid serious overfitting, we reduce the batchsize from 50 to 25. The other hyperparameters are the same as we mentioned in our paper.
>
> |  Method   | Dirichlet-0.6  |  Dirichlet-0.1  |
> |  :----:   |  :----:   |  :----:   |
> |  FedAvg   |  63.56%   |  58.61%  |
> |  SCAFFOLD |  65.11%   |  60.94%  |
> |  FedInit  |  65.08%   |  60.71%  |
>
> Though we do not adjust the coefficient of the relaxed initialization and just use 0.1, the proposed FedInit could still achieve the comparable test accuracy of the SCAFFOLD method. Due to the time limitations of this rebuttal stage, we will fine-tune this parameter in the final version to find the optimal value. However, the improvements of the relaxed initialization are still strong even on the setup of 1000 clients.
>
> |  Method   | Vanilla  |  + relaxed initialization  |  improvement (value) |
> |  :----:   |  :----:   |  :----:   |  :----:   |
> |  FedAvg   |  58.61%   |  60.71% (FedInit)     |  +2.1%  |
> |  SCAFFOLD |  60.94%   |  61.47%   |  +0.53%   |
>
> The vanilla FedAvg could be improved by 2% and the vanilla SCAFFOLD could be improved by 0.53% (Dirichlet-0.1). Likewise, we do not finetune $\beta$ and only select it as 0.1. We believe this improvement will increase further after a simple search. Due to the time limitation, we will complete the entire experiment in the next version.
>
> ## About the question of "How does participation rate influence performance?":
>
> To explore the performance of the different participation rates, we do the following experiments. Due to the time limitation, we test the proposed FedInit on two setups.
>
> #### Total clients=100, CIFAR-10, Dirichlet-0.1, Communication round=500
> |            | 5%  |  10%  |  20%  |  30%  |  50%  |
> |  :----:   |  :----:   |  :----:   |  :----:   |  :----:   |  :----:   |
> | Top-1 Accuracy  |  73.16%  |  75.89%  |  76.04%  |  76.33%  |  76.52%  |
> |  Loss (min)  |  0.5832  |  0.4799  |  0.4040  |  0.3854  |  0.3706  |
>
> #### Total clients=1000, CIFAR-10, Dirichlet-0.1, Communication round=500
> |            | 1%  |  2%  |  5%  |  10%  |
> |  :----:   |  :----:   |  :----:   |  :----:   |  :----:   |
> | Top-1 Accuracy  |  60.61%  |  61.83%  |  62.30%  |  62.12%  |
> |  Loss (min)  |  1.1381  |  1.0839  |  1.0462  |  1.0374  |
>
> The same as above, to avoid overfitting, we adjust the batchsize to 25 on the setup of 1000 clients. The other hyperparameters are fixed the same as we adopted in our paper. Both experiments validate the influence of changing the participation ratios. When the number of active clients are increasing, the performance achieves better. This also matches the theoretical analysis. One key problem is that we should make sure that the training process does not overfit when adopting larger scale of clients. We show the loss curve in the one-page .pdf file to validate their trends.
>
> ### It is a pleasure to discuss this with you, which will help us to improve this work further. We explain and prove the concerns mentioned in the reviews. If there are any questions, we are happy to continue the discussion with you. Thank you again for reading this rebuttal.

---

> > ### Comment · Reviewer_sALG · 2023-08-18
> > **Thanks for the response**
> >
> > I have read all the reviews from other reviewers and responses from the author.  I would like to keep the score.

---

> > > ### Author Response · Authors · 2023-08-18
> > > **Thank you very much for the reviews**
> > >
> > > Dear reviewer sALG, thank you very much for your affirmation of our work. We will add the experiments of total 1000 clients into the Appendix in the next version. Thank you again for this review!

---

### Official Review · Reviewer_9rh8 · 2023-07-05

**Soundness:** 3 good
**Presentation:** 2 fair
**Contribution:** 3 good
**Rating:** 6
**Confidence:** 4

**Summary:**

This paper aims to solve the “client-drift” problem in Federated Learning, which is caused by the NonIID data. Specifically, this paper proposes initializing the local model of each client with its personalized model to alleviate the problem. Further, the paper theoretically analyzes the impact of inconsistency on the convergence of FL. Besides, extensive experiments also demonstrate the effectiveness of the proposed method.

**Strengths:**

1.	The idea of initializing the local model for solving the NonIID problem is interesting.
2.	The theoretical analysis for the proposed method is solid.
3.	The experimental results demonstrate the effectiveness of the proposed method.


**Weaknesses:**

1.	Although the theoretical results are sufficient to verify the convergence of the proposed method, they cannot present the effectiveness of the initialization strategy in principle. More specifically, how does the proposed method reduce the divergence term compared to the vanilla FedAvg?

2.	The workflow of the proposed method seems not correct. Are the locations of line 10 and line 12 placed correctly?

3.	How to obtain the value of $w_{i,k}^{t-1}$ is not clear. The client may not participate in the previous round $t-1$ under the setting of random client selection. If the local model $w_{i,k}^{t’-1}$ is obtained in many previous rounds, the motivation of bias correction using $ w_{i,k}^{t’-1} - w^{t-1}$ is not reasonable due to the significant gap between $t’$ and $t$.

4.	The key hyperparameter K is set inappropriately. In typical FL literature, the number of local epochs is usually set to range from $5$, $10$, and up to $20$, which may contain many local iterations (mini-batches). However, the experiment of this paper only adopts a small number of local iterations instead of local epochs, e.g., 5 local iterations in Table 1, which seems manually adjusted.

5.	It would be better if there is a figure to illustrate the principles of the proposed method.


**Questions:**

See above.

**Limitations:**

Yes.

---

> ### Author Rebuttal · Authors · 2023-08-07
>
> Thank you very much for your review and affirmation of our work. We'll answer your questions one by one in the following, including some misunderstandings and some essential academic questions worth exploring. We are also very honored to share some of our understandings with you.
>
>
> ## About the question of "theoretical results cannot present the effectiveness of the initialization strategy in principle":
>
> Thank you for pointing out this. We discuss our theoretical analysis in each subsection of Section 4 and our theoretical analysis comprehensively explains why FedInit works better than vanilla FedAvg.
>
> Our paper mainly explores the performance of the consistency term in FL to understand how it affects each term. The proposed FedInit method could be considered as a method that adopts a local relaxed initialization on the FedAvg. From the perspective of optimization, under this perturbation, we prove that the order of the convergence rate will not be affected by this perturbation (Thm 1 & 2).
>
> In section 4.3.4, thm.6 indicates that the consistency term mainly affects the generalization rather than optimization in the excess risk. When we consider the PL-condition, the optimization error achieves $\mathcal{O}(1/T)$ and the generalization error achieves $\mathcal{O}(1/T^{\frac{1}{1+cL}})$. The dominant term comes from the consistency term. Therefore, our proof reveals that the consistency term affects the generalization more than the optimization in the vanilla FedAvg method. In Thm.6, we discuss the minimization in the paragraph "Selection of $\beta$". The constant part of the dominant term (related to $\beta$) is $\frac{(1+\beta)^{\frac{1}{\beta cL}}}{\sqrt{1-96\beta^2}}$ (Line.287) where $0<\beta<\frac{\sqrt{6}}{24}$. It can be easily verified that when $\beta$ is selected as a specific positive number the constant term achieves the minimal. This value is less than the limit value of the constant term when $\beta$ approaches 0.
>
> In summary, our proofs mainly indicate:
>
> (1) (**Consistency term mainly affects generalization in FL.**) From the perspective of excess risk (sometimes it could be considered as the test error), the consistency term mainly affects the generalization and dominates the excess risk.
>
> (2) (**Relaxed initialization works better.**) From the excess risk perspective, Selecting a small positive $\beta$ works better than vanilla FedAvg theoretically, which could minimize the constant part in the dominant term. Both experiments and theoretical analysis could validate the efficiency of relaxed initialization.
>
>
> ## About the question "The workflow of the proposed method seems not correct.":
>
> Thank you very much and we are sorry for this mistake. Line.10 should be moved outside the inner loop. After local training, it sends the optimized local state to the global server. And, Line.12 should be moved outside the middle loop. It aggregates the global model after the selected local clients send their local models to the global server. We will fix it.
>
>
> ## About the question of "How to obtain the value of $w_{i,K}^{t-1}$ is not clear.":
>
> Thank you very much for pointing out this. Actually at each communication round $t$, if client $i$ is selected to be active, $w_{i,K}^t$ is trained by $K$ steps. If the client $i$ is not selected, $w_{i,K}^t=w_{i,K}^{t-1}$. The same operation could be found in FedDyn [1], it inherits the states of the local clients who do not participate in the training at the current stage.
>
> Why does it work?
>
> A client may not be selected after many communication rounds, but its last optimized local state still maintains the information of the local minimal. Local training makes each local model close to the local optimum (in most cases, local training could make the local training loss very low). Compared with adopting the current global model as initialization, the effort of the RI is to make the local initialization far away from the local optimum. Therefore, using the historical state is still useful. Though it may not be very accurate, it still works as a correction away from the local optimal solution.
>
> Actually, we submit our code in the Supplementary Material. In the experiments, we just use the above policy to train the FedInit and achieve comparable results as many advanced methods. RI also helps many methods achieve higher test accuracy, i.e. SCAFFOLD and FedSAM. It could work as a light plug-in to further improve their performance as shown in Table.2.
>
> [1] Federated Learning Based on Dynamic Regularization (ICLR 2021)
>
>
> ## About the question of "The key hyperparameter K is set inappropriately.":
>
> We are sorry for this misunderstanding. **The $K$ in the theoretical analysis means the iteration. But we do 5 local epochs in the training. We will correct it in the next version**. We submit the code in the Supplementary Material and it can be checked there we use 5 local epochs for training.
>
> We referred to the experimental setups from the baselines in our paper and chose an intermediate value of 5 to fairly compare their performance. We are sorry for this misunderstanding and we will correct the number of $K$ in our paper. We can use $E = 5$ to represent the local epochs in the hyperparameters introduction.
>
>
> ## About the question of "It would be better if there is a figure to illustrate the principles of the proposed method.":
>
> Thank you very much for pointing out this. Since we took up a lot of space in the introduction of theoretical analysis, we did not provide the schematic diagram in the submitted version. We provide a figure in the one-page .pdf file to illustrate the principles.
>
>
> ### It is a pleasure to discuss this with you, which will help us to improve this work further. We explain and prove the concerns mentioned in the reviews. If there are any questions, we are happy to continue the discussion with you. Thank you again for reading this rebuttal.

---

> > ### Comment · Reviewer_9rh8 · 2023-08-15
> > **Responses to answers**
> >
> > I have read all the reviews from other reviewers and responses from the author. I would like to improve my score.

---

> > > ### Author Response · Authors · 2023-08-15
> > > **Thank you very much for the reviews**
> > >
> > > Dear reviewer 9rh8, we appreciate you for finding the mistakes in the algorithm box and suggesting drawing the schematic to illustrate the principles, which helps us further complete this work. We will fix the mistakes and misunderstandings mentioned above in the next version. Thank you again for the review!

---

### Official Review · Reviewer_qyTU · 2023-07-10

**Soundness:** 3 good
**Presentation:** 3 good
**Contribution:** 3 good
**Rating:** 7
**Confidence:** 4

**Summary:**

This paper proposes an efficient stage-wise initialization for the federated learning paradigm, named FedInit, which could be extended as a plug-in to several existing methods. It provides the theoretical analysis on both the convergence and generalization to illustrate how consistency term affects the FL. Experiments also show its efficiency in practical scenarios.

**Strengths:**

1. FedInit is a light and nimble technique that does not introduce extra communication costs in the FL framework. Compared with other algorithms, FedInit can almost achieve the same performance as SOTA algorithms while maintaining the same communication cost as the vanilla FedAvg. And, Relaxed initialization (RI) could be easily extended into other advanced algorithms as a plug-in and efficiently improve their performance.

2. This paper explores the impacts of global consistency constraints in FL, which is an essential and interesting problem in federated learning. It also indicates the relationships between generalization and optimization.

3. This paper provides a theoretical analysis of the excess risk to comprehensively explain how the dominant factors affect the final performance (test accuracy) throughout the training process. Ablation studies are conducted to validate the conclusion of the theoretical analysis on the hyperparameters, i.e. local interval $K$, coefficient of RI $\beta$.


**Weaknesses:**

1. The relaxed initialization (RI) looks like an extrapolation algorithm between the global model and the local models as $w+\beta(w-w_i)$. Several methods including the benchmarks of this paper, i.e. FedProx, and FedCM, are accustomed to applying interpolation methods to adjust local training to make the entire heterogeneous training smoother. So what is their essential difference? I think the author should add a paragraph to discuss it.

2. This paper involves a lot of analytical proofs, so I suggest that the author write a simple proof sketch for each sub-sections to facilitate reading.

3. In Table.2, RI helps to largely improve the performance of FedSAM and SCAFFOLD but shows limited help for FedCM and FedDyn. What is the main reason for this phenomenon? Does it imply that RI is subject to some existing algorithms? I think the author should add a paragraph to discuss it.

4. Some typos (for example):
(1) Line.150 "... ... its generality"

(2) Line.79 "client drift" and Line.151 "client-drift"

(3) Line.192 the sentence is difficult to read


**Questions:**

Please see the weaknesses.

**Limitations:**

Please see the weaknesses.

---

> ### Author Rebuttal · Authors · 2023-08-07
>
> Thank you very much for your review and affirmation of our work. We'll answer your questions one by one in the following, including some misunderstandings and some essential academic questions worth exploring. We are also very honored to share some of our understandings with you.
>
> ## About the question "What is the essential difference between the extrapolation and interpolation in relaxed initialization (RI)?":
>
> Although they look alike, they have completely opposite effects in relaxed initialization. Here we can express these two formations in a unified form as $w_{init}=w+\beta(w-w_i)$, where $\beta\geq 0$ means extrapolation and $\beta < 0$ means interpolation. When we review the inconsistency due to local updates on the heterogeneous dataset, it means each local client is always optimized to achieve its local optimum. From this perspective, each optimized $w_{i}$ is always close to the local optimum.
>
> Therefore, if we select the interpolation method ($\beta < 0$), from the above formulation, the local initialization state will get close to the local optimum than the vanilla global state, which means each local state overfits more on the local dataset than before. It will introduce a larger inconsistency. However, if we use the extrapolation, it means the local initial state will get far away from the local optimum than the vanilla global state. Since this initialization is used as compensation, the local state obtained after the next stage of training will also get close to each other. We also theoretically prove that using a small positive $\beta$ could reduce the constant part of the generalization error bound. That's why we use extrapolation as our proposed RI calculation.
>
> We also test these experiments in the paper. In Table.3 (Line.348) in our paper,
> |  $\beta$   | -0.2  |  -0.1  |  0 (FedAvg)  |  0.01  |  0.02  |  0.05  |  0.1  |  0.15  |
> |  :----:   |  :----:   |  :----:   |  :----:   |  :----:   |  :----:   |  :----:   |  :----:   |  :----:   |
> |  Accuracy   |  64.70   |  67.47  |  72.53   |  72.82  |  73.45   |  74.65  |  **75.95**   |  44.47  |
>
> Using a positive $\beta$ is better.
>
>
> ## About the question "I suggest that the author write a simple proof sketch for each sub-sections to facilitate reading.":
>
> Thank you for pointing out this. Due to space limitations, we delete the original proof sketch in the submission version. We will add some proof sketches in the next version to further clarify our proof process.
>
>
> ## About the question "What is the main reason that RI shows different performance on different methods?"
>
> The proxy-based methods may not be improved by RI. Because they utilize a proxy term $\Vert w - w_{Init}\Vert^2$ in the local objectives at each communication round, which are sensitive to the initialization state. For instance, FedDyn [1] uses a dynamic regularization term to correct the local objective constantly, which could achieve the same objective for different clients. It is a variant of ADMM which controls the dual variables. Due to the impact of the dual variable, if we directly use the relaxed initial state, the vanilla dual variable will not match the proxy term. In other words, it requires a more specific study on this ADMM-type method in FL. FedCM [2] uses the global update as a biased correction to update the local training, which has achieved a very high consistency. That is why the performance of the (RI + FedCM) is weak. The promotion for enhancing the consistency of FedCM from RI is limited by the correction term.
>
> [1] Federated Learning Based on Dynamic Regularization
>
> [2] FedCM: Federated Learning with Client-level Momentum
>
>
>
> ## About the question of "Some typos":
>
> Line.150, we will fix it as "its generalization performance".
>
> Line.151, we will fix it as "client drift".
>
> Line.192, this sentence says that $G$ controls the heterogeneity level, and the local interval $K$ is limited by the $G$.
>
>
>
> ### It is a pleasure to discuss this with you, which will help us to improve this work further. We explain and prove the concerns mentioned in the reviews. If there are any questions, we are happy to continue the discussion with you. Thank you again for reading this rebuttal.

---

### Author Rebuttal · Authors · 2023-08-07

**We are very grateful to all the reviewers for their valuable comments.**

We make individual responses to each reviewer to address the concerns they raised. Here we submit the one-page .pdf file which contains some experiment curves mentioned by review sALG and a figure to illustrate the principle of the proposed relaxed initialization technique which is mentioned by review 9rh8 and g4w5. Thanks again for the valuable comments of the reviewers.

---

### Decision · Program_Chairs · 2023-09-21

**Decision:**

Accept (poster)

**Comment:**

The authors have developed a stage-wise relaxed initialization-based federated learning algorithm. In this approach, the local model at the start of each local training stage is initialized by moving away from the current global model in the opposite direction. This algorithm aims to address the challenge of client drift. When the initialization parameter $\beta$ is sufficiently small, the authors show that the proposed algorithm can discover stationary points or global optimal solutions for federated learning problems when the loss function is nonconvex or satisfies the PL condition, respectively. They have also conducted an analysis of the uniform stability and excess risk of this algorithm.

The theoretical analysis is straightforward, albeit with strong assumptions about the Lipschitz continuity of the loss function. Also,  it is worth noting that the paper does not mention communication efficiency.

All reviewers have provided positive feedback on the quality of this work, and most of them acknowledged authors' responses. Additionally, there were no responses to my initiated discussions during the AC-reviewer discussion phase. Based on my evaluation, it appears to be a borderline work.